# KITAB: Evaluating LLMs on Constraint Satisfaction for Information Retrieval

**Marah I Abdin**[1]     **Suriya Gunasekar**[1]     **Varun Chandrasekaran**[2]     **Jerry Li**[1]
**Mert Yuksekgonul**[3]     **Rahee Ghosh Peshawaria**[1]     **Ranjita Naik**[1]     **Besmira Nushi**[1]
[1]Microsoft Research, [2]University of Illinois Urbana-Champaign, [3]Stanford University

## Abstract

We study the ability of state-of-the art models to answer *constraint satisfaction* queries for information retrieval (e.g., "a list of `ice cream` shops in `San Diego`"). In the past, such queries were considered to be tasks that could only be solved via web-search or knowledge bases. More recently, large language models (LLMs) have demonstrated initial emergent abilities in this task. However, many current retrieval benchmarks are either saturated or do not measure constraint satisfaction. Motivated by rising concerns around factual incorrectness and hallucinations of LLMs, we present KITAB, a new dataset for measuring constraint satisfaction abilities of language models. KITAB consists of book-related data across more than 600 authors and 13,000 queries, and also offers an associated dynamic data collection and constraint verification approach for acquiring similar test data for other authors. Our extended experiments on GPT4 and GPT3.5 characterize and decouple common failure modes across dimensions such as *information popularity*, *constraint types*, and *context availability*. Results show that in the absence of context, models exhibit severe limitations as measured by irrelevant information, factual errors, and incompleteness, many of which exacerbate as information popularity decreases. While context availability mitigates irrelevant information, it is not helpful for satisfying constraints, identifying fundamental barriers to constraint satisfaction. We open source our contributions to foster further research on improving constraint satisfaction abilities of future models. [1]

## 1 Introduction

Answering factual queries is one of the many emerging abilities of large language models (LLMs) (OpenAI, 2023; Touvron et al., 2023; Chowdhery et al., 2022). This has reinvented the way search engines operate by involving conversational LLMs directly as part of the user experience (e.g., BingChat). As with many emerging abilities, rigorous evaluation remains a challenge due to a continuous lack of appropriate benchmarks, benchmark saturation, training data contamination, and difficulties in evaluating open-ended generated output from LLMs (Chang et al., 2023; Liang et al., 2022). At the same time, several concerns have arisen over repeated occurrences of LLMs fabricating false information or providing irrelevant content (informally termed as hallucinations) (Bender et al., 2021; Bommasani et al., 2021; Zhang et al., 2023; Sun et al., 2023a).

This work studies and evaluates constraint satisfaction capabilities of LLMs in the context of information retrieval (IR). Similarly to traditional constrained search problems (Meseguer, 1989), constraint satisfaction queries in IR are queries that include a set of constraints to be satisfied by the generated output. The framework has been recently proposed for studying and detecting factual errors of LLMs by Yuksekgonul et al. (2023) as a useful perspective which also connects information popularity and constraint feasibility to the LLM's ability to satisfy such constraints. Here, we employ the same framework to guide LLM evaluation and experimental design. Queries with constraints can also be considered as the more general form of keyword, boolean, or pattern-matching queries (Baeza-Yates et al., 1999) and faceted web search (Tunkelang, 2009; Hahn et al., 2010),

---

[1]https://huggingface.co/datasets/microsoft/kitab
[†]Correspondence to marah.abdin@microsoft.com and besmira.nushi@microsoft.com.

| | Irrelevant information ↓ | Relevant information (Books from the author) | | Completeness ↑ | All Correct ↑ |
|---|---|---|---|---|---|
| | | Satisfied ↑ | Unsatisfied ↓ | | |
| **GPT4** | 0.26 \| 0.33 \| 0.00 | 0.51 \| 0.49 \| 0.78 | 0.24 \| 0.19 \| 0.21 | 0.24 \| 0.26 \| 0.70 | 0.08 \| 0.08 \| 0.31 |
| **GPT3.5** | 0.20 \| 0.44 \| 0.00 | 0.44 \| 0.26 \| 0.68 | 0.36 \| 0.30 \| 0.32 | 0.16 \| 0.16 \| 0.47 | 0.07 \| 0.02 \| 0.15 |

Table 1: Aggregated model performance on KITAB for 3 prompts NO-CONTEXT | SELF-CONTEXT | WITH-CONTEXT (see definitions in § 3.2) for queries requesting a list of books from a given author satisfying one additional book constraint. Both models have high rates of irrelevant information and poor constraint satisfaction across the board. Context availability mitigates irrelevant information rate, but constraint satisfaction still remains low. Full correctness (i.e., perfect match of the post-processed model output and the ground truth) is strikingly low across all conditions and models but there is visible improvement for WITH-CONTEXT. Similar results for queries with two book constraints are shown in Appendix, Table 5.

where constraints are expressed in natural language. For example, the query "A list of research papers authored by {author} published after {year}" naturally specifies at least two constraints on the required output. While the variety of constraint types across user requests in an LLM-powered search engine can be large and some constraints may be more difficult to parse and verify, fundamentally, many user interactions fall under this definition, particularly in scenarios where users seek specific and precise information rather than open-ended, creative text.

While several benchmarks and reports exist for evaluating factual correctness on simple, single constraints queries that expect a single-output item (e.g., "Which city is the capital of Ukraine") (Lin et al., 2021; Elazar et al., 2021; Kwiatkowski et al., 2019; Zellers et al., 2019), many of them have saturated. Little is understood about the performance of LLMs on more complex queries with several constraint types and that generate longer outputs. Staying consistent with constraints on longer text generations is important to study as this is a major differentiator between previous and newer architectures (Chang et al., 2023), which exhibit better self-consistency. Surprisingly, as we will show in this analysis, staying consistent with external constraints remains challenging even for state-of-the-art LLMs (GPT4 and GPT3.5) trained on internet-scale data (see Table 1). To better understand how and when these inconsistencies occur, we contribute KITAB, a dataset and dynamic data collection approach focused on literature queries, as a classical example of a domain that can benefit from efficient retrieval and has sufficient public information potentially also used during training (e.g., on Wikipedia). KITAB queries are of the form: "A list of all books from Toni Morrison published between 1970-1980?", where the first constraint is fixed to an author and the following can vary among lexical, temporal, and named entity constraints.

We use KITAB to test LLMs across different controlled conditions, specifically, their: i) baseline ability to retrieve all books from an author (ALL-BOOKS), ii) performance on queries that have both an author constraint and book constraints using only the parametric knowledge (NO-CONTEXT), iii) performance when they have access to a "complete" context with all books from the author, to differentiate between parametric and retrieval-augmented settings (WITH-CONTEXT), and finally iv) performance for standard chain-of-thought prompts and prompts that require the LLM to first construct its own context with all books from the author, as a self-sufficient retrieval approach that does not use other systems (SELF-CONTEXT). These conditions enable us to carefully characterize and decouple failure modes for the task, and draw insights as follows:

- Using only their parametric knowledge, state-of-the art LLMs have a high rate of presenting irrelevant (potentially hallucinated) books not written from the given author, varying between 12% and 41%. Irrelevant information increases abruptly for authors with lower popularity.
- Complete context availability addresses irrelevance, but constraint satisfaction failures remain a major obstacle across both LLMs and different constraint types, even with complete context.
- Self-retrieval approaches significantly *increase* the rate of irrelevant (potentially hallucinated) information and fabricated titles that are not from the author, for the sake of satisfying constraints.
- While GPT4 improves all scores when compared to GPT3.5, the difference between the two LLMs is not as dramatic showing that scale alone may not address the filtering with constraints problem. All correctness (i.e., perfect match with the ground truth) remains notably lower than 35%.

Besides the dataset and a detailed report on GPT4 and GPT3.5, the work also contributes an approach for collecting and cleaning other versions of KITAB using the same process but on a disjoint author list. The process can be of significant importance to confront benchmark saturation or leakage, and to support independent testing in situations when the initial dataset may be used in training.

## 2 BACKGROUND & RELATED WORK

**Factual Queries:** Most prior work focuses on locating specific facts in the LLM's parameters (Meng et al., 2022; Geva et al., 2023; Mallen et al., 2022), or understanding how the LLM's performance in these tasks can be improved (Chuang et al., 2023). While these works indirectly benchmark the LLM's ability to correctly respond to factual queries, they primarily focus on short responses, using datasets that are saturated (i.e., with reasonably high or SOTA performance already), or worse–contaminated. For example, Nori et al. (2023) note that GPT4 is able to reproduce questions from SQuAD 2.0 (Rajpurkar et al., 2018) verbatim, while OpenAI (2023) notes contamination for MMLU (Hendrycks et al., 2020), and Sun et al. (2023b) highlight how GPT4 achieves SOTA results for BEIR (Thakur et al., 2021).

A promising solution to fact-finding failures and hallucinations is to combine generation with retrieval mechanisms as done in retrieval augmented generation (RAG) (Nakano et al., 2021; Lewis et al., 2020)). As we discuss in § 3.2, we simulate this setting by providing the desired complete information in-context and then evaluate the LLM in its ability to respond to factual queries. In practice, (pre-)retrieval in RAG can introduce new challenges across many domains, especially when the retrieval engine is unreliable or expensive.

**Constraint Satisfaction:** As discussed by Yuksekgonul et al. (2023), many queries (and tasks) can be viewed through the lens of constraint satisfaction. Using this same lens provides us with a natural framework for generating queries with varying notions of complexity i.e., by altering the constraints. The main distinction between this study and work by Yuksekgonul et al. (2023) is that we contribute a dataset (and functional evaluation) that is challenging even for large proprietary models like GPT4; Yuksekgonul et al. (2023) propose an attention-based method for mechanistic understanding and detecting failures of open-source models using model internals. More broadly, other tasks that can be viewed as constraint satisfaction problems include planning (Valmeekam et al., 2022), instruction tuning (Zhou et al., 2023), and controlled generation (Zheng et al., 2023).

**Constraint and Query Complexity**: One way of measuring query complexity is using the notion of *constrainedness* (Meseguer, 1989; Gent et al., 1996), which views this as a function of the number of solutions for a given constraint. In similar spirit, we measure the complement of the ratio between the number of solutions $S$ that satisfy the constraint and the total number of items in the domain $N$ (higher constrainedness, more complex), i.e., $\kappa = 1 - \frac{S}{N}$. Constrainedness can also be seen as the opposite of query *selectivity* in database systems (Getoor et al., 2001), i.e., the percentage of records that satisfy the query. Constraint *popularity* measures the popularity of entities within specific constraints (more popular, less complex). Ideally, popularity would directly measure information frequency in training data. In absence of such information, we use the number of sitelinks in the author's WikiData page. In many open-world problems, it is not possible to directly compute popularity or constrainedness, which is why we make this information available in KITAB.

## 3 METHOD

**Research Questions.** Whether users may be looking up general knowledge facts (e.g., "Which vaccines are due at `four years old`?") or using LLMs to research and collect information on a topic (e.g., "A list of all authors from `Africa` who have won the `Nobel Prize`?"), failure to satisfy the given constraints and factual errors may lead to lack of trust, frustration, and safety concerns (e.g., in settings such as soliciting healthcare advice). Our goal is to dissect model performance and create transparency around when and how current LLMs fail on constraint satisfaction queries. To guide dataset and experimental design, we focus on the following research questions:

**RQ1:** How does model performance vary depending on the *type* of constraint?

**RQ2:** How does model performance change if *complete information is made available in-context*?

**RQ3:** How does model performance vary depending on *content popularity and constrainedness*?

**RQ4:** What are the main bottlenecks in constraint satisfaction queries in IR for current LLMs?

To answer these questions, we designed the KITAB dataset. KITAB contains queries with high diversity in the (i) type of constraints, (ii) number of candidate solutions (i.e., constrainedness), and

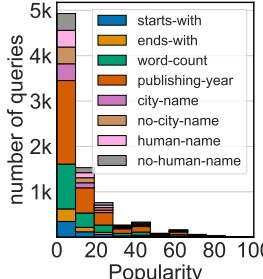

Figure 1: Author popularity for one book constraints.

|  | One book constraints | | Two book constraints | |
|---|---|---|---|---|
| **Constraint Type** | **# queries** | **constrainedness** | **# queries** | **constrainedness** |
| starts-with | 598 | 0.90 | 2163 | 0.92 |
| ends-with | 482 | 0.89 | 1782 | 0.91 |
| word-count | 1672 | 0.53 | 1630 | 0.81 |
| human-name | 611 | 0.77 | 292 | 0.89 |
| no-human-name | 611 | 0.23 | 801 | 0.78 |
| city-name | 611 | 0.92 | 197 | 0.81 |
| no-city-name | 611 | 0.08 | 831 | 0.77 |
| publishing-year | 3043 | 0.80 | 1804 | 0.89 |
| Summary | 8239 | 0.67 | 4750 | 0.87 |

Table 2: KITAB statistics on constraint frequency and average constrainedness. Two book constraint queries have more than one constraint type.

(iii) author popularity (i.e., a proxy for frequency in the dataset). Figure 1 and Table 2 summarize main data statistics. More detailed information is also available in Appendix, Figure 5 and 6.

### 3.1 KITAB DATA COLLECTION

**Author sampling.** To seed the data collection, we first sample 20,000 authors (i.e., entities marked as writers) randomly from WikiData, as a public data source that has been potentially used for the training of several models (Gao et al., 2020). To avoid potentially inaccurate data and extreme outliers, we filter out authors that were born before 1850 and those that have less than 10 or more than 300 works linked to their profile, which results to 1505 authors. Next, we cross-reference these authors with the Open Library repository using the author name and year of birth, keeping only those that have at least five works in Open Library (after book cleaning and deduplication), resulting in 599 authors. These filtering choices ensure that the final sample contains both a useful and natural distribution of author popularity for which it is possible to construct satisfiable queries. We focus on popularity since previous works (Carlini et al., 2022; Biderman et al., 2023; Yuksekgonul et al., 2023; Mallen et al., 2022) identified popularity as a key factor for factual errors. While Mallen et al. (2022) measure popularity through the number of page visits, Shokouhi (2011) demonstrated that page visits are seasonal and might paint a false picture of popularity. Henceforth, similar to Yuksekgonul et al. (2023), we use the number of website links in WikiData as a proxy to information popularity. Figure 1 shows the distribution of the number of site-links in WikiData (as a proxy for popularity) across the whole sample, which includes an additional control set of 12 handpicked well-known authors from the five continents. The control set was used for repeated quality checks on the data cleaning workflow described next. The final sample contains 611 authors.

**Book collection.** Using the name of the author and their year of birth, we cross-reference the Open Library corpus and collect all books from the author that are tagged to be in English by the API, or where the language field is empty. Then, we make an additional check using the Azure Cognitive Services Language API for language detection such that we keep only the earliest English edition titles, given that our prompts are also in English. Further, the data cleaning process involves a number of quality and consistency checks, namely on deduplication and cross-checking the authorship and publication year of the book on both the Open Library and WikiData. We also keep variants of the same title to facilitate model evaluation when the same book may be known with slightly different titles and bylines (e.g., "Gödel, Escher, Bach" vs. "Gödel, Escher, Bach: An Eternal Golden Braid"). Despite our best efforts in collecting a complete and accurate set of books, we also faced a variety of challenges in retrieval and cleaning, which we further describe in Appendix C.1. To estimate the extent of which potential data cleaning issues may impact the data quality of KITAB and further evaluation, we also undertook a manual data annotation exercise during which we searched the web for titles provided by GPT4 and GPT3.5 but that were marked as "not from the author" in our dataset. In summary, we find that based on a manual annotation of a subsample of queries, less than 5% of the queries to GPT4 and less than 6% of the queries to GPT3.5 may potentially be affected by cases where the model finds a book title that is not in KITAB (and that will consequentially be marked as not from the author during our evaluation). While this can be remediated by using further data sources, the impact of missing information on our evaluation is minor.

Together with books, KITAB also provides a variety of book *metadata* to enable verification functions for constraint satisfaction, including: publication year, list of human or city names in the title (if any). Entity recognition for human names was done using both Azure Cognitive Services and

GPT4 (Template 4 in Appendix D), as we found the two approaches to be complementary for detecting names from different cultures. For city names, we use Azure Cognitive Services along with Geonames, a database of cities with more than 1000 inhabitants (Opendatasoft, 2023).

**Constraints and queries.** All queries in KITAB have the following form:

```
List all books written by Toni Morrison (born in 1931) that
                             author constraint
were first published between 1970-1980.
          book constraint
```

In each query, the first constraint is always fixed to an author and the following can vary among *lexical* (title starts or ends with a letter, word count in title), *temporal* (published between start and end year), and *named entity* (city or human name present or not present in title) book constraints to test for different constraint satisfaction capabilities. Since there exists a large number of constraint instances depending on their cardinality, we subsample from the large set of queries in a way that ensures i) a balanced representation across constraint types, and ii) a variety of constraints that have different constrainedness. We also add "unsatisfiable" constraints, which do not match any book titles in our data–this constitutes 7.99% of the queries.

The final dataset contains 8239 queries with one book constraint and 4750 queries with two book constraints. Table 2 shows how these queries are distributed across different constraint types. For most double-constraint queries, both constraints are individually satisfiable and generated by combining our single constraint data. Only 0.76% of the queries are jointly unsatisfiable across both constraints. Further details on the constraint sampling process are presented in Appendix § C.2.

To enable offline model evaluation, KITAB includes a mapping of all books that satisfy each of the 12,989 queries. Altogether, this provides a convenient tool also for the evaluation of LLM generated output, which we detail in § 4.1. While for this work we focus on the literature domain, the workflow design can prove useful for other domains as well (e.g., movies, restaurants, research papers etc.).

## 3.2 EXPERIMENTAL CONDITIONS

To answer the presented research questions, we lay out the following experimental conditions that map to specific prompt templates, which are detailed in Appendix D. All templates in this list (except Template 1) ask the model to provide a (prior) brief reason to why a book in the output list satisfies a given constraint, as a standard chain-of-thought approach.

**ALL-BOOKS** (Template 1): List all books from the author. This condition enables us to estimate an upper bound of model performance in retrieving relevant information for all queries, regardless of other constraints. In experimental results, we will use the notion of the ratio of books that are not from the author as the rate of irrelevant information since these items are irrelevant to the query, regardless of whether the other constraints are satisfied. This condition then helps in decoupling how information irrelevance changes between queries that have none, one, or two adittional book constraints, for settings that use only the model's parametric knowledge.

**NO-CONTEXT** (Template 2a): List all books from the author that also satisfy other book constraints. The same template is used for testing two book constraints. This condition will measure model performance in satisfying different types of constraints, using only the model's parametric knowledge.

**WITH-CONTEXT** (Template 2b): First, provide a full list of books from the author as input context to the model. Then, ask the model to list all books from the author that also satisfy another book constraint. The same template is used for testing two book constraints. This condition intends to simulate retrieval-augmented settings Nakano et al. (2021); Lewis et al. (2020) where the retrieval part of the system can provide a complete context to the model and the model's task is then to just run and verify the constraints. While retrieved context may often also be incomplete in practice, here we provide the list of all books from the author known to KITAB to isolate potential failures to only model shortcomings for verifying constraints. Note that some of the constraints (but not all) could also be solved through declarative languages (i.e., SQL) if the input context is structured or one could even require the model to write code for constraint verification. However, given the broader nature of our queries and the fact that relevant input context is usually not structured, here we are interested in testing the native abilities of the model to verify basic constraints.

SELF-CONTEXT (Template 3): Ask the model to first self-retrieve all books from the author, and then use that list to find those that also satisfy book constraints. This tests whether the model can simulate a self-sufficient retrieval setting, as a more advanced chain-of-thought approach.

SINGLE-ITEM (Template 4): Ask the model to apply a constraint on a single book title to decouple the performance of the model in applying constraints on a single item from applying constraints to a whole list. Here, we sample 400 queries using a single book as described in Appendix § C.2.

## 4 EXPERIMENTS

We evaluate the performance of GPT4 and GPT3.5 on our dataset, with prompt templates and maximum token length as defined in Section 3.2. All experiments were done with temperature 0.

### 4.1 METRICS AND EVALUATION

The guiding principle for the design of metrics used in this evaluation was to be as lenient as possible to the model while still being able to measure important positive and negative trends. In early evaluations we found that model answers may vary slightly from the ground truth answer, e.g., by omitting a byline in the title, outputting variations of the title, or repeating a title. To ensure these factors do not artificially decrease model performance, we design our metrics to accommodate for such partial and/or fuzzy matches. For counting constraints, we also consider titles that have one word more or less than the specified constraint as satisfied, to add more tolerance to the evaluation. Surprisingly, even with all of this leeway, SOTA models still perform poorly on KITAB.

**Calculating information irrelevance and partial satisfaction.** For each query and the answer that the model provides, we calculate the fraction of irrelevant books, as well as the fraction of satisfying and unsatisfying answers, in a way which accommodates for repeated titles, partial titles, and fuzzy matches. We do so as follows. First, we process the final list of answers from the model into a set of $n$ strings $K = \{k_1, \ldots, k_n\}$. For each $k_i$, we check if there exists a book in the ground truth set of books by that author which is either a string subset match for $k_i$ (in both directions), or if any book in the ground truth is at 80% match in Levenshtein distance. If it passes either of these checks, we associate it to that ground truth solution. Otherwise, we mark the book as irrelevant (i.e., not from the author). We then cluster all strings which match to the same ground truth into a single cluster. This process yields a partition of $K$ into $m$ clusters $C_1, \ldots, C_m$ where each cluster is either a size 1, containing a single irrelevant book (i.e., a book that is not written by the author), or a cluster where all books are mapped to the same ground truth book. We call the former the set of irrelevant clusters, and the latter the relevant clusters. We then further break down the relevant clusters into two types. We say that a relevant cluster is a satisfying cluster if any of the strings in the cluster satisfy the constraint, and otherwise we say it is an unsatisfying cluster. Note that intentionally, we are not naming irrelevant clusters as hallucinations because it can be the case that a book retrieved by the LLM exists but is not from the author. This is more difficult to check because it requires access to the whole set of books ever written, albeit qualitatively we see several cases with numerous titles that do not even appear on web search and potentially do not exist.

With these definitions, we can now define our metrics. For each query, we report the fraction of irrelevant, satisfying, and unsatisfying clusters. We denote these three quantities by $p_{\text{irr}}$, $p_{\text{sat}}$, and $p_{\text{unsat}}$, respectively. By definition, $p_{\text{irr}} + p_{\text{sat}} + p_{\text{unsat}} = 1$. We emphasize that these are very generous terms for the model, and that as a result, it is quite possible that we are overestimating the true model performance. However, we believe that this makes our qualitative finding that SOTA models still struggle on this task to be even more interesting.

**Calculating completeness and all-correctness.** We also wish to evaluate the fraction of correct answers that the model returns, i.e., its completeness. For every query, we define the completeness of the model's answer as follows. For each book in the ground truth, we check if it is an approximate match to a book by the model, using the same methodology as above (i.e. subset matching and fuzzy matching). We then define the completeness of the model's answer, denoted $p_{\text{comp}}$, to be the fraction of ground truth answers that have such an approximate match. Finally, we say that the model's answer is all correct if $p_{\text{sat}} = 1$ and $p_{\text{comp}} = 1$. This is the strictest evaluation metric that measures whether the model made no factual errors for the query and found all relevant information.

| | Single Item | Irrelevant information ↓ | | | Relevant information (Books from the author) Satisfied ↑ | | | Unsatisfied ↓ | | | Completeness ↑ | | | All Correct ↑ | | |
|---|---|---|---|---|---|---|---|---|---|---|---|---|---|---|---|---|
| **starts-with** | 0.96 | 0.41 | 0.36 | 0.01 | 0.50 | 0.57 | 0.79 | 0.09 | 0.07 | 0.20 | 0.29 | 0.31 | 0.83 | 0.11 | 0.17 | 0.47 |
| **ends-with** | 0.80 | 0.23 | 0.38 | 0.00 | 0.23 | 0.28 | 0.31 | 0.54 | 0.34 | 0.69 | 0.15 | 0.17 | 0.46 | 0.04 | 0.05 | 0.06 |
| **word-count** | 0.58 | 0.21 | 0.33 | 0.00 | 0.61 | 0.53 | 0.63 | 0.17 | 0.14 | 0.37 | 0.07 | 0.09 | 0.39 | 0.00 | 0.00 | 0.02 |
| **human** | 0.70 | 0.36 | 0.39 | 0.01 | 0.41 | 0.46 | 0.84 | 0.23 | 0.14 | 0.15 | 0.16 | 0.19 | 0.61 | 0.06 | 0.07 | 0.23 |
| **no-human** | 0.65 | 0.32 | 0.36 | 0.00 | 0.57 | 0.55 | 0.90 | 0.10 | 0.09 | 0.10 | 0.25 | 0.31 | 0.83 | 0.00 | 0.00 | 0.13 |
| **city** | 0.56 | 0.12 | 0.46 | 0.00 | 0.77 | 0.38 | 0.66 | 0.11 | 0.16 | 0.34 | 0.33 | 0.26 | 0.38 | 0.31 | 0.20 | 0.31 |
| **no-city** | 0.54 | 0.36 | 0.34 | 0.00 | 0.59 | 0.61 | 0.93 | 0.05 | 0.05 | 0.07 | 0.31 | 0.32 | 0.91 | 0.00 | 0.00 | 0.26 |
| **pub-year** | 1.00 | 0.21 | 0.27 | 0.00 | 0.46 | 0.47 | 0.90 | 0.33 | 0.26 | 0.10 | 0.31 | 0.34 | 0.88 | 0.11 | 0.12 | 0.53 |
| **Summary** | 0.80 | 0.26 | 0.33 | 0.00 | 0.51 | 0.49 | 0.78 | 0.24 | 0.19 | 0.21 | 0.24 | 0.26 | 0.70 | 0.08 | 0.08 | 0.31 |

Table 3: GPT4 performance on KITAB for NO-CONTEXT | SELF-CONTEXT | CONTEXT across different constraint types for queries with one book constraint. Results for GPT3.5 are shown in Appendix, Table 4. Similar evaluations for queries with two book constraints are presented in Appendix, Table 6 and 7, respectively.

## 4.2 RESULTS

**Overall results.** We present the overall statistics averaged over the entire dataset in Table 1. For each metric, results are shown for NO-CONTEXT | SELF-CONTEXT | WITH-CONTEXT conditions in order. Overall, GPT4 performs quite poorly on this dataset, and although it performs better than GPT3.5, the difference is not so dramatic, suggesting that improvement on constraint satisfaction tasks may not come simply by scaling up. While chain-of-thought helps improve accuracy, it does not seem sufficient by itself, see Appendix F (Example 1), and in fact, advanced chain-of-thought (measured by SELF-CONTEXT) increases the incidence of irrelevant books. We also observe that while the incidence of irrelevant books becomes negligible when the context is provided (WITH-CONTEXT), this does not solve issues with constraint satisfaction, completeness and all correctness, see Appendix F (Example 2). Model performance remains unreliable even with provided complete context from KITAB, simulating search-assisted settings.

We also break down performance by query type in Table 3 for GPT4 and Appendix, Table 4 for GPT3.5. We find interesting variations between query types. GPT4 struggles much more with ends-with than with starts-with queries. Differently from the starts-with constraint, for the model to satisfy the ends-with ones, it has to plan ahead and look into the future of several token generations that may lead to a sequence ending with a letter. For entity-based queries, we see that negation queries (e.g., doesn't contain) are easier to satisfy and that is reflected in model performance. Yet, even in the best performing types, GPT4 makes a non-negligible fraction of errors.

**Popularity.** We next consider the correlation between popularity (as measured by WikiData sitelinks) and model performance, in Figure 2 for GPT4. See Appendix, Figure 7b for GPT3.5. Surprisingly, while irrelevant information decreases with higher popularity, we do not see a clear positive correlation between popularity and desirable outcomes such as the satisfaction, completeness, and all-correctness. Again, this result shows that constraint satisfaction remains a difficult task to solve only with larger data (i.e., higher popularity). One interesting and, to our knowledge, novel observation is that it seems there is a relatively sharp "phase transition" in the incidence of irrelevant books relative to popularity. When the number of sitelinks for the author is very small, i.e. between 0-10, irrelevance is quite high. Afterwards, the rate of irrelevant books drops, but quickly flattens out, and does not improve with more sitelinks, with any statistical significance. We conjecture that this is because "pragmatic decisions" need to be made during training time; with models devoting memorization resources only after seeing the author a number of times. Of course, this is a simplistic view to the observed quick transition in popularity, and the phenomenon warrants future research. Importantly, all correctness remains strikingly low across all conditions and popularity bins ($< 35\%$). The finding has important implications to the reliability and completeness of information, if models evaluated in this work were to be used as part of larger automated systems.

**Constrainedness.** Figure 3 shows the relationship between constrainedness (as defined in Section 2) and GPT4 model performance. Similar results are shown for GPT3.5 in Appendix, Figure 8b. Here, we see a more nuanced phenomenon when results are aggregated across different constraint types, with model performance resembling an S-curved, almost bimodal distribution, consistent for both models. This is easier to observe in Figure 3 for the WITH-CONTEXT condition, in particular for completeness and all-correctness. To better understand the dynamics, we then disaggregate the same figures but per each constraint type in Appendix, Figures 9 and 10. First, we find that while

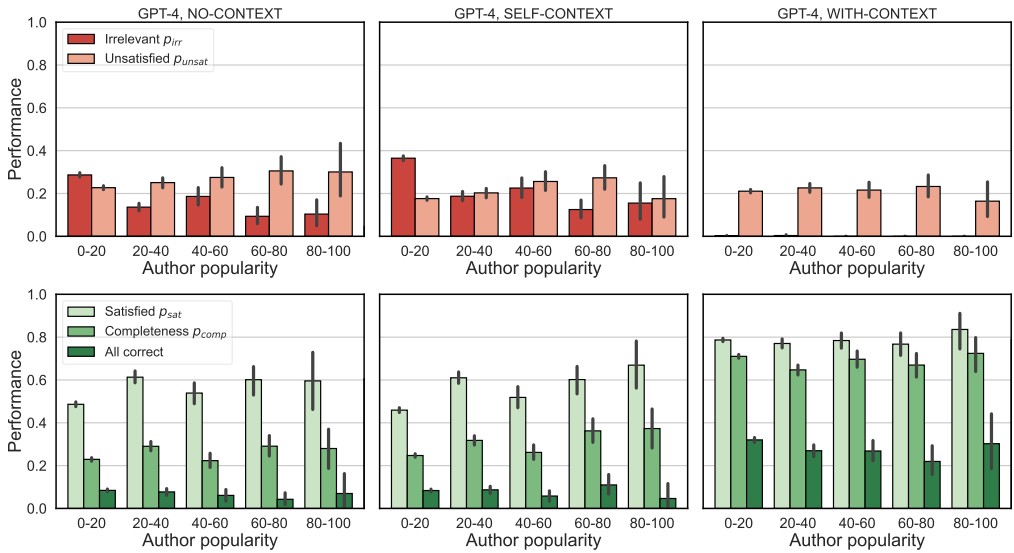

Figure 2: GPT-4 performance on KITAB comparing NO-CONTEXT(left), SELF-CONTEXT(middle) and WITH-CONTEXT(right) queries across various popularity bins. We show trends for irrelevant information, and unsatisfaction rate in top plot; and for satisfaction, completion and correctness rates in the bottom plot.

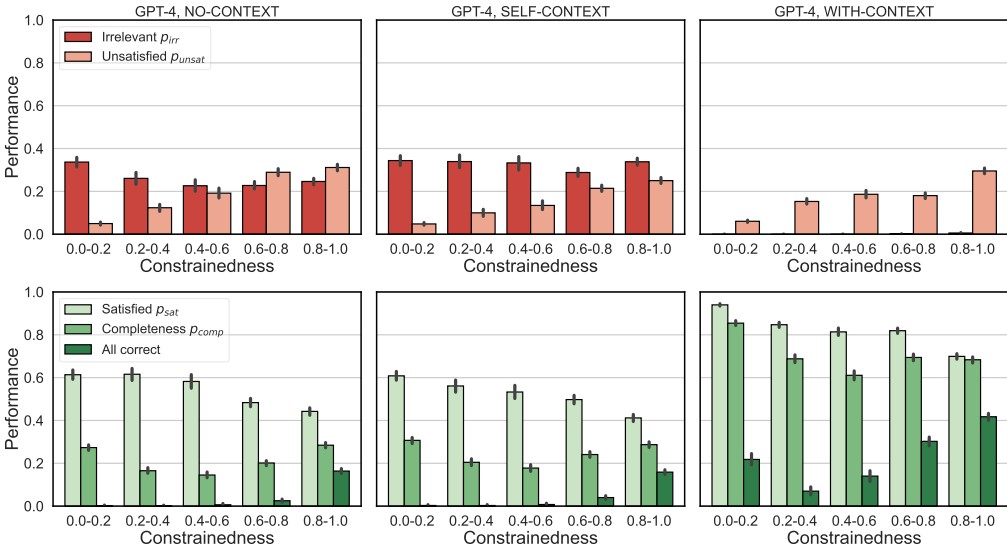

Figure 3: GPT-4 performance on KITAB for queries across various constrainedness bins. Similar to Figure 2, we compare NO-CONTEXT(left), SELF-CONTEXT(middle) and WITH-CONTEXT(right) with irrelevant information and unsatisfaction rates in the top; and satisfaction, completeness, and all correctness rates in the bottom.

for most constraint types a higher constrainedness is related to lower model performance (consistent with findings by Yuksekgonul et al. (2023)), for particular constraints like ends-with and city-name, the opposite is true. In addition, for entity constraints (human and city names) the two forms (entity exists or does not exist in the title) are placed in two different ends of constrainedness. This can also be seen in Table 2 and Figure 6 where negation queries are placed in the lower end of the graph. Thus, when summed up, the overall dynamics can resemble an almost bimodal effect of constrainedness on performance. While we do not have a full explanation to why the ends-with and city-name constraints behave differently, the variation highlights the importance of controlled, large-scale datasets such as KITAB in measuring emergent behavior of LLMs at scale.

**Multiple constraints.** Figure 4 shows model performance on queries with only an author constraint vs. with additional one and two book constraints. Unsurprisingly, model performance consistently decreases for more complex and more constrained queries with two book constraints. As a naïve

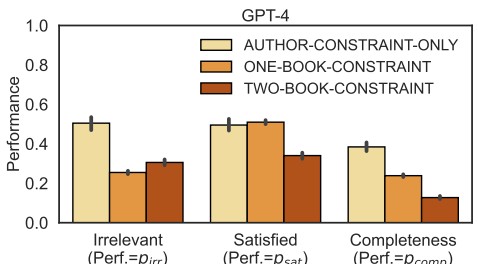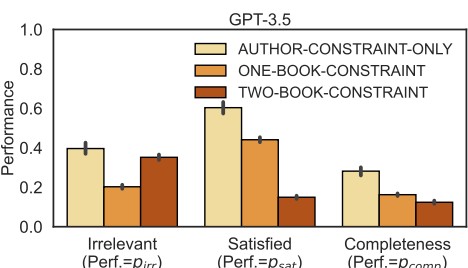

Figure 4: Model performance on queries with only an author constraint vs. plus one book constraint, and plus two book constraints. Results for queries with book constraints are based of NO-CONTEXT (Template 2a).

baseline, we also compare with performance on queries with only the author constraint. While completeness and constraint satisfaction decrease in the direction of no book constraints to two book constraints, irrelevant information follows different dynamics. In particular, models seem to fabricate significantly more irrelevant information when they are asked to list all books from an author. In fact, if one considers the whole set of books by all authors available in the training data as the domain for the ALL-BOOKS queries, the constrainedness of such a query when no other constraints are present is quite high. This may demonstrate that estimating the domain cardinality for computing constrainedness is not straightforward and that some leading constraints (i.e., the author in our case) may serve as conditioning handlebars to the domain size used by the model. The finding however warrants future experimentation for studying if and how such conditioning happens. Further detailed results on model performance by constraint type for queries with two book constraints can be found in Tables 6 and 7 for GPT4 and 3.5.

**Further decoupling analysis.** To better understand how irrelevant information propagates at different stages of our queries, we study the SELF-CONTEXT condition in further detail. We observe that irrelevance for the first part of the chain-of-thought process when the model outputs all books from the author is notably high, 0.42 for GPT4 and 0.47 for GPT3.5. Even though after applying constraints, irrelevance decreases to 0.33 and 0.44, this still remains higher than other conditions as the model is not able to recover from the initial fabricated titles. Qualitatively, we observe that sometimes models collect irrelevant books in condition SELF-CONTEXT such that they can satisfy the constraint later on (see Examples 3 and 4 in Appendix F).

Finally, we look at model performance in satisfying constraints for SINGLE-ITEM lists of books. Here, we measure the accuracy of the model in detecting whether a constraint is satisfied for one title using the same prompt as for WITH-CONTEXT. Model accuracy for SINGLE-ITEM is shown in the first columns of Tables 3 and 4. When comparing these metrics with satisfaction rates from WITH-CONTEXT, we see that constraint types have two very different behaviors consistent across both models. Constraints like starts-with, ends-with, and publication year are easier to check for individual titles than for lists. Instead, entity constraints become easier for lists of book titles, which resonates with the fact that entity recognition is considered a core ability of LLMs on longer text[2].

## 5    CONCLUSION

We presented KITAB, a dataset and dynamic data collection approach for evaluating abilities of large language models to filter information using constraints. The dataset provides convenient flexibility for controlling the type and complexity of constraints in queries that expect longer lists of outputs, beyond simple facts. An in-depth analysis of GPT4 and GPT3.5, two state-of-the-art models deployed in the real-world as part of conversational search systems, showed that despite exciting emerging abilities of such models in finding information, important limitations remain when models fabricate irrelevant information when only parametric knowledge is used or when they fail to satisfy specified constraints even when provided with the most complete and relevant context to filter upon. We hope that the dataset and methodology paves an avenue for future rigorous and large-scale evaluations of emergent abilities in information retrieval problems.

---

[2]We exclude the word-count constraint from this discussion since our evaluation WITH-CONTEXT tolerates answers that are one word longer or shorter than the given constraint.

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

## A  KITAB STATISTICS

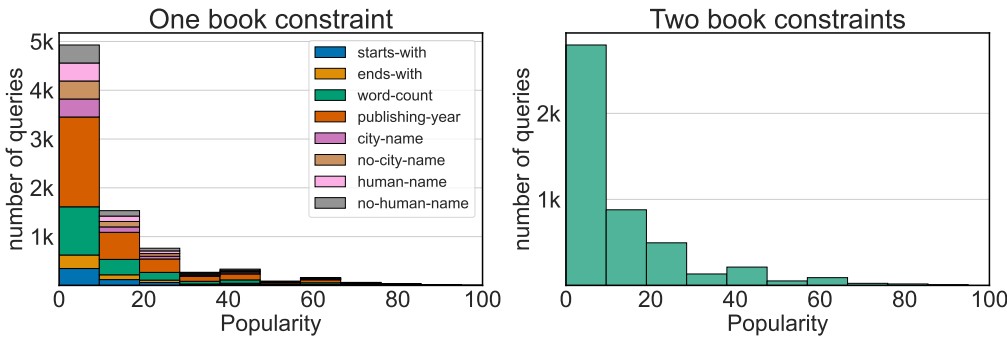

Figure 5: Distribution of queries across author popularity as measured by the number of sitelinks on Wikidata, for queries with a single book constraint (left) and two book constraints (right).

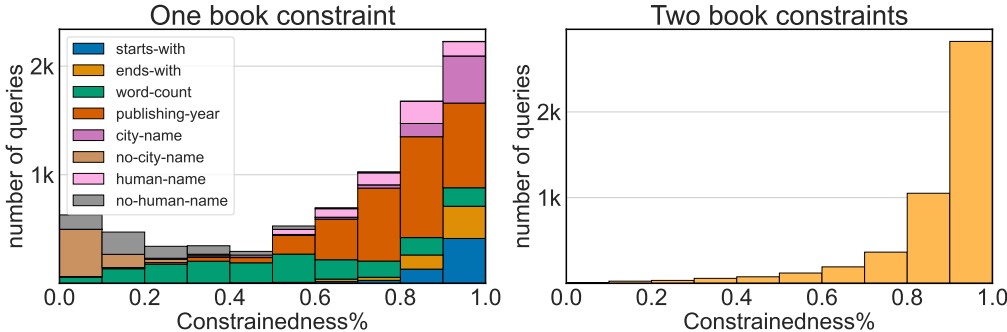

Figure 6: Distribution of queries across author constrainedness as measured by the complement of the ratio between the number of books that satisfy the book constraints and the total number of books from the author. Distribution is shown for queries with a single book constraint (left) and two book constraints (right). Note that most of the distribution in the lower range of constrainedness is dominated by constraints that require no human name or no city name in the title, which are naturally easier to satisfy.

## B  EVALUATION ON QUERIES WITH ONE AND TWO BOOK CONSTRAINTS

| | Single Item | Irrelevant information ↓ | | | Relevant information (Books from the author) Satisfied ↑ | | | Unsatisfied ↓ | | | Completeness ↑ | | | All Correct ↑ | | |
|---|---|---|---|---|---|---|---|---|---|---|---|---|---|---|---|---|
| **starts-with** | 0.83 | 0.33 | 0.47 | 0.01 | 0.36 | 0.35 | 0.80 | 0.32 | 0.18 | 0.19 | 0.13 | 0.18 | 0.49 | 0.04 | 0.09 | 0.22 |
| **ends-with** | 0.59 | 0.18 | 0.42 | 0.00 | 0.15 | 0.07 | 0.16 | 0.67 | 0.51 | 0.83 | 0.12 | 0.07 | 0.34 | 0.03 | 0.00 | 0.02 |
| **word-count** | 0.55 | 0.17 | 0.43 | 0.00 | 0.41 | 0.22 | 0.43 | 0.42 | 0.34 | 0.57 | 0.06 | 0.04 | 0.19 | 0.00 | 0.00 | 0.02 |
| **human** | 0.70 | 0.30 | 0.46 | 0.00 | 0.24 | 0.28 | 0.67 | 0.46 | 0.26 | 0.33 | 0.18 | 0.14 | 0.50 | 0.04 | 0.01 | 0.12 |
| **no-human** | 0.60 | 0.26 | 0.48 | 0.00 | 0.58 | 0.42 | 0.86 | 0.16 | 0.09 | 0.14 | 0.12 | 0.24 | 0.71 | 0.00 | 0.00 | 0.04 |
| **city** | 0.56 | 0.21 | 0.52 | 0.00 | 0.38 | 0.12 | 0.58 | 0.41 | 0.37 | 0.42 | 0.28 | 0.09 | 0.38 | 0.19 | 0.03 | 0.29 |
| **no-city** | 0.50 | 0.23 | 0.48 | 0.02 | 0.72 | 0.48 | 0.91 | 0.06 | 0.03 | 0.07 | 0.11 | 0.26 | 0.79 | 0.00 | 0.00 | 0.13 |
| **pub-year** | 0.92 | 0.17 | 0.41 | 0.00 | 0.52 | 0.23 | 0.84 | 0.32 | 0.35 | 0.15 | 0.22 | 0.22 | 0.54 | 0.12 | 0.03 | 0.23 |
| **Summary** | 0.69 | 0.20 | 0.44 | 0.00 | 0.44 | 0.26 | 0.68 | 0.36 | 0.30 | 0.32 | 0.16 | 0.16 | 0.47 | 0.07 | 0.02 | 0.15 |

Table 4: GPT3.5 performance on KITAB for NO-CONTEXT | SELF-CONTEXT | CONTEXT across different constraint types for queries with one book constraint.

| | Irrelevant information ↓ | Relevant information (Books from the author) | | Completeness ↑ | All Correct ↑ |
|---|---|---|---|---|---|
| | | Satisfied ↑ | Unsatisfied ↓ | | |
| **GPT4** | 0.31 \| 0.00 | 0.34 \| 0.54 | 0.35 \| 0.46 | 0.13 \| 0.52 | 0.06 \| 0.19 |
| **GPT3.5** | 0.35 \| 0.01 | 0.15 \| 0.40 | 0.50 \| 0.60 | 0.12 \| 0.36 | 0.00 \| 0.07 |

Table 5: Aggregated model performance on KITAB for NO-CONTEXT | CONTEXT for queries with two book constraints.

| | Irrelevant information ↓ | Relevant information (Books from the author) | | Completeness ↑ | All Correct ↑ |
|---|---|---|---|---|---|
| | | Satisfied ↑ | Unsatisfied ↓ | | |
| **starts-with** | 0.37 \| 0.00 | 0.36 \| 0.58 | 0.27 \| 0.41 | 0.17 \| 0.63 | 0.11 \| 0.29 |
| **ends-with** | 0.28 \| 0.00 | 0.22 \| 0.39 | 0.50 \| 0.61 | 0.09 \| 0.46 | 0.05 \| 0.15 |
| **word-count** | 0.28 \| 0.00 | 0.41 \| 0.57 | 0.30 \| 0.43 | 0.08 \| 0.42 | 0.04 \| 0.16 |
| **human** | 0.32 \| 0.00 | 0.29 \| 0.52 | 0.40 \| 0.48 | 0.10 \| 0.46 | 0.05 \| 0.17 |
| **no-human** | 0.26 \| 0.00 | 0.41 \| 0.62 | 0.33 \| 0.38 | 0.11 \| 0.51 | 0.03 \| 0.16 |
| **city** | 0.57 \| 0.00 | 0.11 \| 0.42 | 0.32 \| 0.58 | 0.02 \| 0.12 | 0.01 \| 0.09 |
| **no-city** | 0.31 \| 0.00 | 0.40 \| 0.61 | 0.29 \| 0.38 | 0.15 \| 0.58 | 0.03 \| 0.16 |
| **pub-year** | 0.28 \| 0.00 | 0.32 \| 0.54 | 0.40 \| 0.46 | 0.14 \| 0.58 | 0.06 \| 0.16 |
| **Summary** | 0.31 \| 0.00 | 0.34 \| 0.54 | 0.35 \| 0.46 | 0.12 \| 0.52 | 0.06 \| 0.19 |

Table 6: GPT4 performance on KITAB for NO-CONTEXT | CONTEXT across different constraint types for queries with two book constraints. The type needs to appear at least once in the query to be grouped under a constraint type. Satisfaction rate is reported jointly for both constraints (i.e., both need to be satisfied).

| | Irrelevant information ↓ | Relevant information (Books from the author) | | Completeness ↑ | All Correct ↑ |
|---|---|---|---|---|---|
| | | Satisfied ↑ | Unsatisfied ↓ | | |
| **starts-with** | 0.42 \| 0.01 | 0.13 \| 0.49 | 0.44 \| 0.50 | 0.13 \| 0.43 | 0.00 \| 0.12 |
| **ends-with** | 0.29 \| 0.00 | 0.07 \| 0.22 | 0.64 \| 0.78 | 0.08 \| 0.32 | 0.00 \| 0.05 |
| **word-count** | 0.35 \| 0.01 | 0.18 \| 0.39 | 0.47 \| 0.60 | 0.13 \| 0.28 | 0.00 \| 0.05 |
| **human** | 0.31 \| 0.00 | 0.10 \| 0.32 | 0.59 \| 0.68 | 0.10 \| 0.35 | 0.00 \| 0.03 |
| **no-human** | 0.29 \| 0.00 | 0.25 \| 0.45 | 0.46 \| 0.55 | 0.11 \| 0.38 | 0.00 \| 0.05 |
| **city** | 0.44 \| 0.01 | 0.05 \| 0.20 | 0.51 \| 0.79 | 0.07 \| 0.21 | 0.00 \| 0.02 |
| **no-city** | 0.31 \| 0.00 | 0.25 \| 0.48 | 0.44 \| 0.51 | 0.10 \| 0.44 | 0.00 \| 0.05 |
| **pub-year** | 0.38 \| 0.01 | 0.15 \| 0.45 | 0.48 \| 0.55 | 0.16 \| 0.37 | 0.00 \| 0.07 |
| **Summary** | 0.35 \| 0.01 | 0.15 \| 0.40 | 0.50 \| 0.60 | 0.12 \| 0.36 | 0.00 \| 0.07 |

Table 7: GPT3.5 performance on KITAB for NO-CONTEXT | CONTEXT across different constraint types for queries with two book constraints. The type needs to appear at least once in the query to be grouped under a constraint type. Satisfaction rate is reported jointly for both constraints (i.e., both need to be satisfied).

## C  FURTHER DETAILS ON KITAB DATA COLLECTION

### C.1  DATA CLEANING

Here, we detail all steps involved in the data cleaning of KITAB. These steps are also available in our dynamic data collection for future users of KITAB, who may reproduce the same workflow for other author samples.

**Book filtering.** We only retrieve books that are tagged to be in English by Open Library and those that have no assigned language, since all our prompts are in English and require the model to return only English titles. We keep the books that have no assigned language to improve book collection completeness. However, we still check through the Azure Cognitive Services API whether these titles are in English before adding them to the list. Next, we also remove books that appear to have more than two authors since many of such books will not necessarily appear as books from a particular author. In most cases, these are collections of works amongst many authors (e.g., "The Best Short Stories 2022: The O. Henry Prize Winners"). Finally, we also found a list of titles in Open Library that were part of an author's collection but not written by the author. To mitigate this, we cross checked with Wikidata for the same title and made sure that the author on Wikidata

matches the one on OpenLibrary. This step is commonly used during data integration for cleaning purposes (Dong & Rekatsinas, 2018) and significantly improved the data quality overall.

**Deduplication.** To deduplicate potentially redundant book titles, we first lower case and strip all punctuation from title strings and then cluster the books retrieved via Open Library by using i) fuzzy matching [3] with 80% threshold, and ii) subset checks for cases when the book title may appear in a longer or abbreviated form (e.g., "Gödel, Escher, Bach" vs. "Gödel, Escher, Bach: An Eternal Golden Braid"). We specifically remove the first word of a title if the title starts with "The" or "A/An" for deduplication purposes, and apply the same strategy during model evaluation itself so that we do not penalize the model if it did or did not miss a title simply because of these minor details. During deduplication, we keep as a publishing year the minimum value of publishing years across all titles in the cluster. Note that the deduplication process also affects the list of ground truth books in KITAB, as the same book may appear multiple times on Open Library. To alleviate this, we keep track of redundant titles of the same book during the book collection stage and then employ the redundant titles as well while computing query constrainedness and completeness. For instance, in the example above, if both variants of the book title are present in the Open Library ground truth (e.g., "Gödel, Escher, Bach" vs. "Gödel, Escher, Bach: An Eternal Golden Braid"), the book itself would be merged into one and marked as satisfying both constraints: "title ends with the letter h" and "title ends with the letter d".

**Manual checks for quality.** The manual checks included looking at two samples of 200 queries for each model and their respective output and inspecting titles that the model had mentioned but we had marked them as irrelevant information (i.e., not from the author). For these titles, we searched on the Web to see if there exists a book from that author with the same title. The process identified that 5% and 6% of the queries in the GPT4 and GPT3.5 samples respectively had at least one title in the model output that indeed belongs to the author according to web search but that title is not present on OpenLibrary. Note that the impact of this variance on irrelevant information rates is in fact much lower than 5% because it is measured across queries and not individual books, as an upper bound estimate.

## C.2 SAMPLING CONSTRAINTS AND QUERIES

Further, we describe the construction and selection of KITAB's constraint satisfaction queries and ground truth.

**Queries with one book constraint.** For each author, we generate constraints by associating every title with starting and ending letters, word count, a 5-year publication year range, and whether the title has human or city names. From this large accumulation, we then randomly sample 15% of the constraints for title beginnings or endings and 50% for title word count. We also append unstaisfiable constraints of each type, for instance, we randomly select letters that no book title in our set starts with. The set of unsatisfiable queries constitutes 7.99% of the queries. The final set of single book constraints is 8239 queries.

**Queries with two book constraints.** For all double-constraint queries, both constraints are individually satisfiable and generated by combining our single constraint data. Only 0.76% of the queries are jointly unsatisfiable across both constraints. For a variation in difficulty, we isolate more easily satisfiable constraints that combine title starts with and published in year-range that map to more than one book and sample 25% of the latter; of the remaining double constraints, we sample 5% that map to a single book and 10% that map to more than 2 books. The final set has 4750 queries.

**SINGLE-ITEM queries.** From KITAB's one book constraint data, we randomly select 200 queries, sampling 50 queries each for title starts and ends with constraints, 30 each for published within a range and word count in title, and 20 each for the presence of a human or city name in the title. For each query, we randomly select a single book item satisifying the constraint from the ground truth and provide it as context. Additionally, for every satisfiable constraint query, we also sample a book by the author that is not present in the ground truth as a juxtapose context, resulting in a total of 400 single item queries.

---

[3] https://github.com/seatgeek/fuzzywuzzy

# D PROMPTING TEMPLATES

---

**[TEMPLATE 1 ALL-BOOKS]: List all books from the author.**
Maximum token length = 1000

```
List of all books written by {$author} (born in {$birth_year}).  All book
                             ‾‾‾‾‾‾‾‾‾‾‾‾‾‾‾‾‾‾‾‾‾‾‾‾‾‾‾‾‾‾‾‾‾
                                      author constraint
titles need to be in English.  Always finish your response with the
following format, do not add any additional text or comments:
Output:
1. Title:  <title>
2. Title:  <title>
...
N. Title:  <title>
```

---

**[TEMPLATE 2A NO-CONTEXT]: List all books from the author that satisfy other constraints, no context.**
Maximum token length = 400

```
List of all books written by {$author} (born in {$birth_year}) satisfying
                             ‾‾‾‾‾‾‾‾‾‾‾‾‾‾‾‾‾‾‾‾‾‾‾‾‾‾‾‾‾‾‾‾
                                     $author constraint
all the following criteria.  All book titles need to be in English.
Think step-by-step.  Give a 1-2 sentence reason for why the books
satisfy the criteria.  Criteria:  {$constraints} Remember that every
                                  ‾‾‾‾‾‾‾‾‾‾‾‾‾‾
                                  book constraints
book in the output list needs to satisfy all the criteria.  Always
finish your response with the following format.  Do not add any
additional text or comments after the output list.
Output:
1. Reason:  <reason>.  Title:  <title>
2. Reason:  <reason>.  Title:  <title>
...
N. Reason:  <reason>.  Title:  <title>
```

---

**[TEMPLATE 2B WITH-CONTEXT]: List all books from the author that satisfy other constraints, with context.**
Maximum token length = 1000 (200 for the SINGLE-ITEM condition)

```
The following is a list of books by {$author} (born in {$birth_year})
                                    ‾‾‾‾‾‾‾‾‾‾‾‾‾‾‾‾‾‾‾‾‾‾‾‾‾‾‾‾‾‾‾
                                              $author constraint
with publication dates in parenthesis.  List:
{$all_books}
‾‾‾‾‾‾‾‾‾‾
Find all books in this list that satisfy all the following criteria.
Think step-by-step.  Give a 1-2 sentence reason for why the books
satisfy the criteria.  Criteria:  {$constraints} Remember that every
                                  ‾‾‾‾‾‾‾‾‾‾‾‾‾‾
                                  book constraints
book in the output list needs to satisfy all the criteria.  Always
finish your response with the following format.  Do not add any
additional text or comments after the output list.
Output:
1. Reason:  <reason>.  Title:  <title>
2. Reason:  <reason>.  Title:  <title>
...
N. Reason:  <reason>.  Title:  <title>
```

---

**[TEMPLATE 3 SELF-CONTEXT]: List all books from the author that satisfy one other constraint, self-retrieve context.**

Maximum token length = 3000

```
List of all books written by {$author} (born in {$birth_year}) satisfying
                              └──────────────────────────┘
                                     $author constraint
all the following criteria.  All book titles need to be in English.
Criteria:  {$constraints} First, retrieve all books by {$author} (born
           └───────────┘
            book constraints
in {$birth_year}) and list them in the "All Books" list.  Then, select
the subset of books that satisfy Constraint 1 and list them under the
"Final Output" list.  Think step-by-step.  Give a 1-2 sentence reason
for why the books satisfy the criteria.  Remember that every book in the
final output list needs to satisfy all the criteria.  Always finish your
response with the following format.  Do not add any additional text or
comments after the output list.
All Books:
1. Title:  <title>
2. Title:  <title>
...
N. Title:  <title>
Final Output:
1. Reason:  <reason>.  Title:  <title>
2. Reason:  <reason>.  Title:  <title>
...
N. Reason:  <reason>.  Title:  <title>
```

---

**[TEMPLATE 4 NAME-CHECK]: Find all books that contain a human name in the title.**

```
The following is a list of books.  List:
{$all_books}
Find all books that contain a human name in the title.  Always finish
your response with the following format.  Do not add any additional text
or comments after the output list.
Output:
1. Reason:  <reason>.  Title:  <title>
2. Reason:  <reason>.  Title:  <title>
...
N. Reason:  <reason>.  Title:  <title>
```

# E ADDITIONAL RESULTS ON PERFORMANCE RELATION TO POPULARITY AND CONSTRAINEDNESS

## E.1 POPULARITY

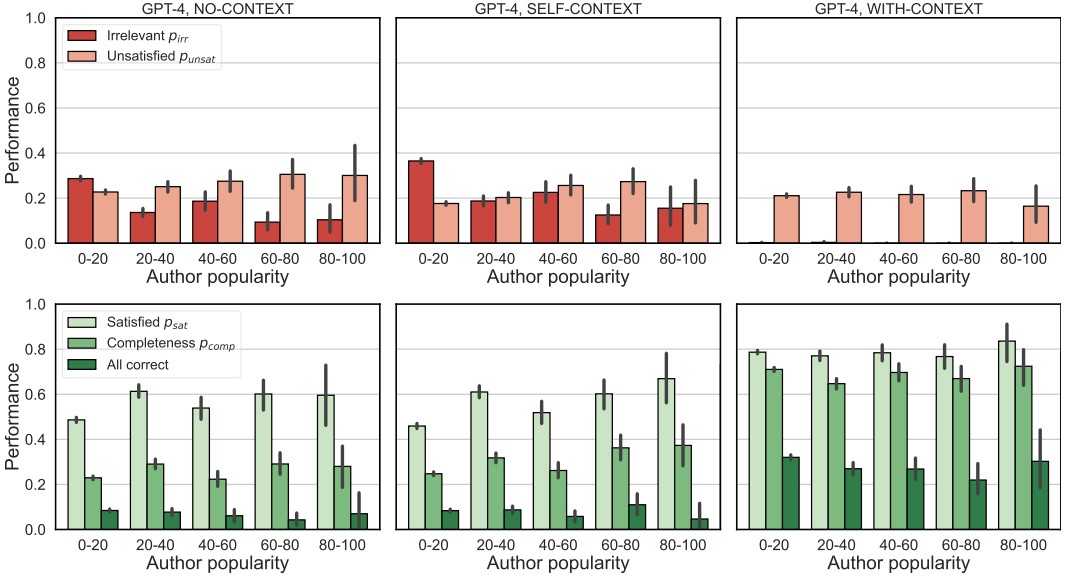

(a) GPT-4 performance on KITAB comparing NO-CONTEXT(left), SELF-CONTEXT(middle) and WITH-CONTEXT(right) queries across various popularity bins. We show trends for irrelevant information, and unsatisfaction rate in top plot; and for satisfaction, completion and correctness rates in the bottom plot.

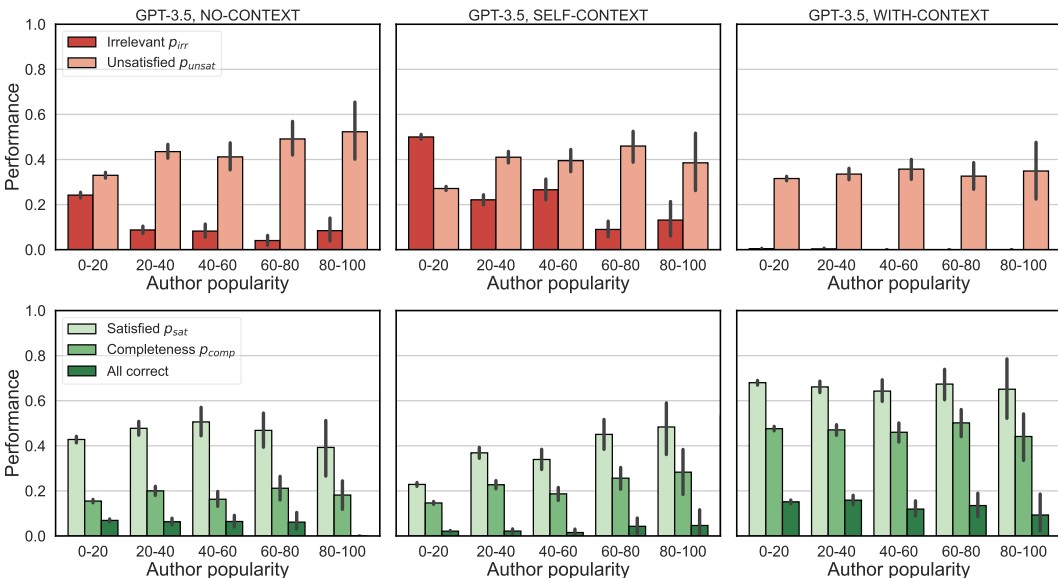

(b) GPT-3.5 performance on KITAB comparing NO-CONTEXT(left), SELF-CONTEXT(middle) and WITH-CONTEXT(right) queries across various popularity bins. We show trends for irrelevant information, and unsatisfaction rate in top plot; and for satisfaction, completion and correctness rates in the bottom plot.

Figure 7: GPT-4 and GPT-3.5 performance vs. popularity.

## E.2 CONSTRAINEDNESS

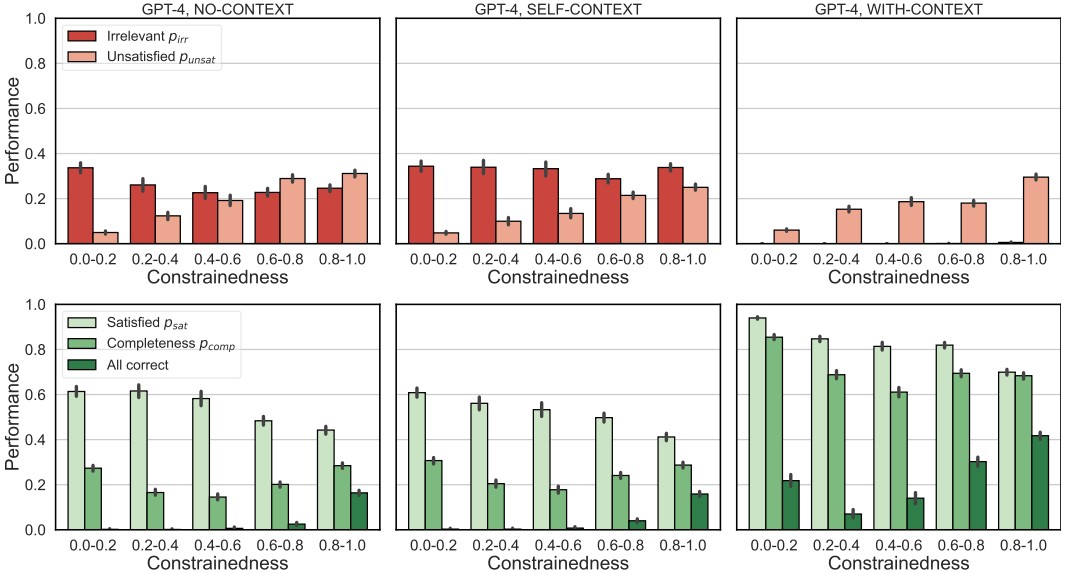

(a) GPT-4 performance on KITAB comparing NO-CONTEXT(left), SELF-CONTEXT(middle) and WITH-CONTEXT(right) queries across various constrainedness bins. We show trends for irrelevant information, and unsatisfaction rate in top plot; and for satisfaction, completion and correctness rates in the bottom plot.

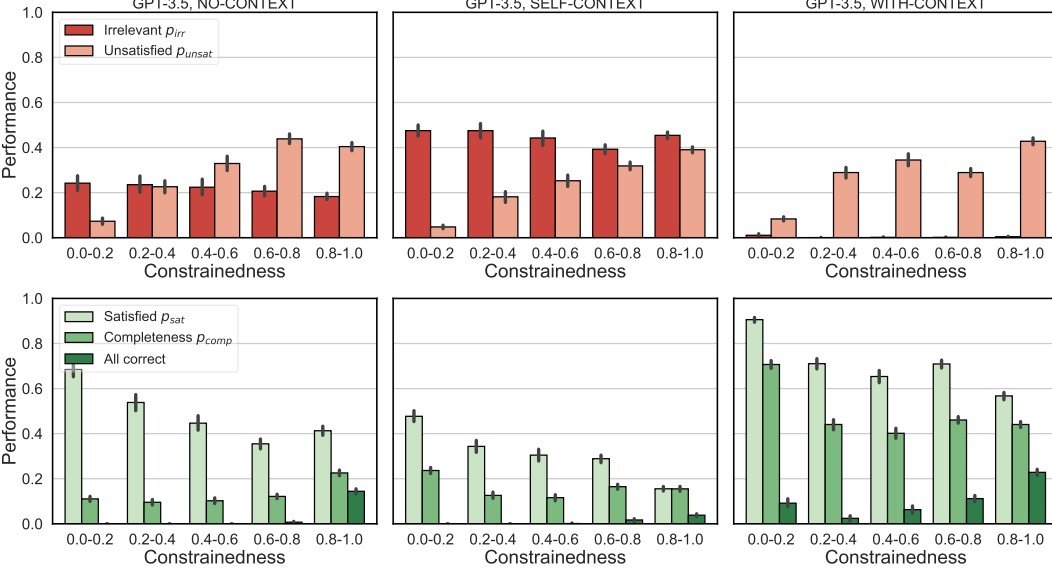

(b) GPT-3.5 performance on KITAB comparing NO-CONTEXT(left), SELF-CONTEXT(middle) and WITH-CONTEXT(right) queries across various constrainedness bins. We show trends for irrelevant information, and unsatisfaction rate in top plot; and for satisfaction, completion and correctness rates in the bottom plot.

Figure 8: GPT-4 and GPT-3.5 performance vs. constrainedness.

## E.3 BREAKDOWN OF GPT-4 PERFORMANCE BY CONSTRAINT TYPE

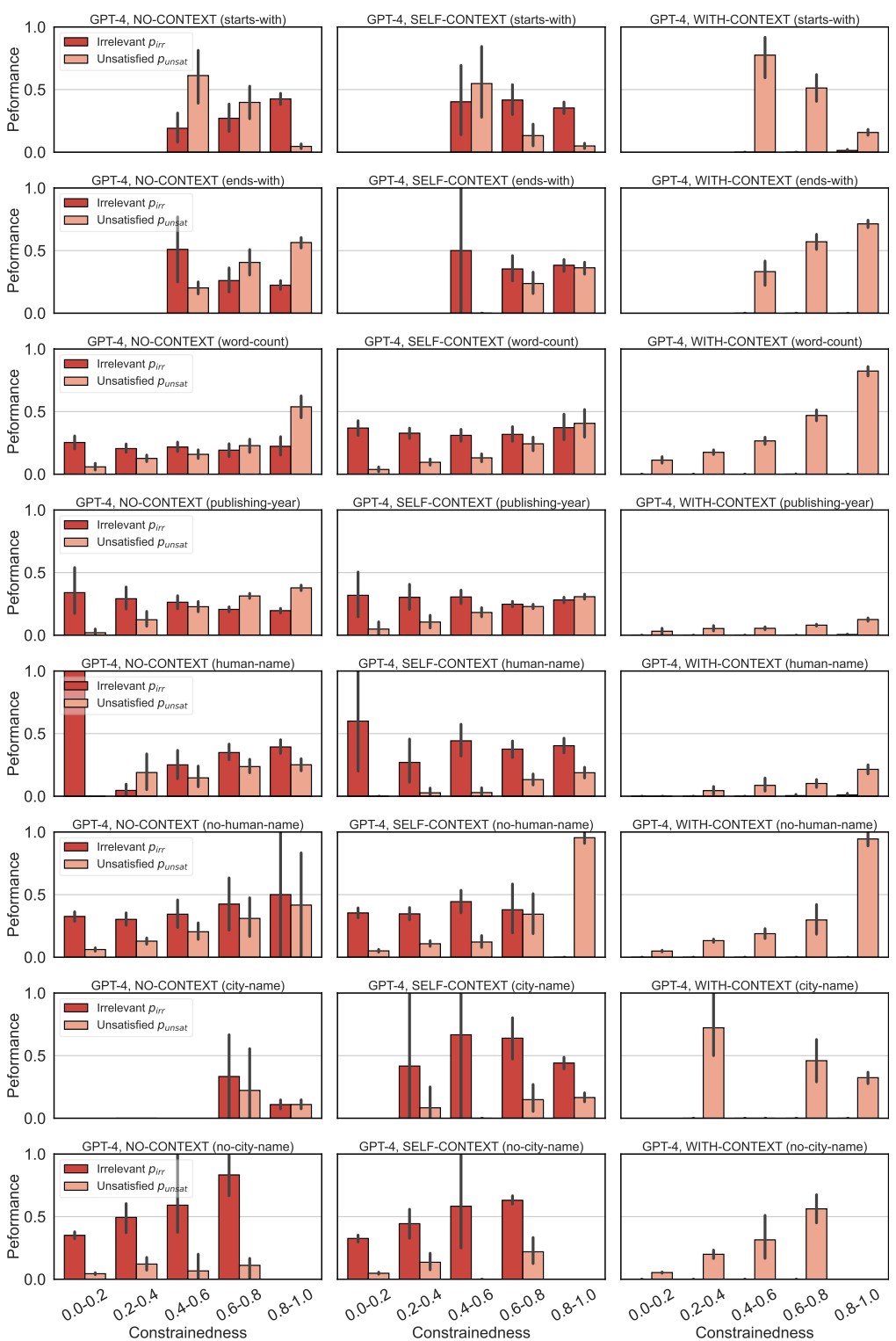

Figure 9: Constrainedness vs irrelevance and unsatisfaction for GPT-4 across constraint types.

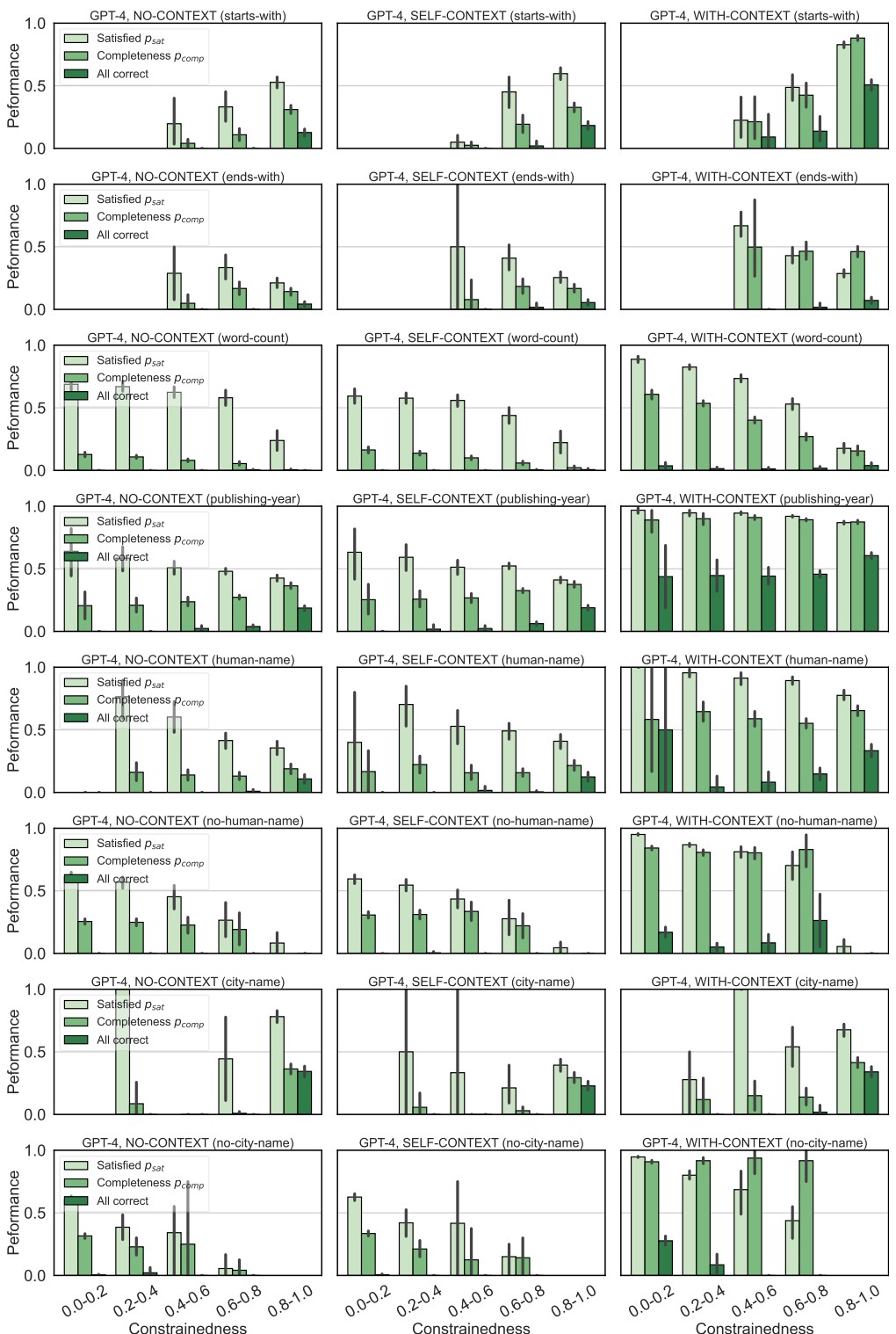

Figure 10: Constrainedness vs satisfaction, completeness and correctness for GPT-4 across constraint types.

## F EXAMPLES

---

**[EXAMPLE 1]: High unsatisfaction rate with NO-CONTEXT.**

**Author**: Michael Scott
**Constraint**: Book title contains only 4 words.
**KITAB ground truth**: ``[A Celtic odyssey', 'Celtic Wisdom for Business', 'Green and golden tales', 'Irish animal tales.', 'Irish folk and fairy tales', 'Irish ghosts and hauntings', 'Irish hero tales', 'Judith and spider', 'Magical Irish folk tales', 'The Childrenof Lir', 'The Culai heritage', 'The book of Celtic wisdom', 'The last ofthe fianna', 'The quest ofthe sons', 'The river gods', 'The seven treasures', 'The thirteen hallows]''
**KITAB all books**" ``['A Celtic odyssey (1985)', 'Billy the Kid and the vampyres of Vegas (2011)', 'Celtic Wisdom for Business (2001)', 'Earthlord (1992)', 'Firelord (1994)', 'Gemini game (1993)', 'Green and golden tales (1988)', 'Irish Fairytales (Green and Golden Tales) (1989)', 'Irish animal tales. (1989)', 'Irish folk and fairy tales (1983)', 'Irish ghosts and hauntings (1994)', 'Irish hero tales (1989)', 'Judith and spider (1992)', 'Lottery (1993)', 'Magical Irish folk tales (1995)', 'Mirror image (2016)', 'October moon (1992)', 'Reflection. (1992)', 'Tales from the land of Erin (1985)', 'The Childrenof Lir (1986)', 'The Culai heritage (2001)', 'The Magician (2008)', 'The alchemyst (2007)', 'The book of Celtic wisdom (2002)', 'The enchantress (2012)', 'The last ofthe fianna (1987)', 'The necromancer (2010)', 'The quest ofthe sons (1988)', 'The river gods (1991)', 'The seven treasures (1992)', 'The thirteen hallows (2011)', 'The warlock (2011)', 'Windlord (1991)']''
**GPT4 books**: ``['alchemyst the secrets of the immortal nicholas flamel', 'magician the secrets of the immortal nicholas flamel', 'necromancer the secrets of the immortal nicholas flamel', 'warlock the secrets of the immortal nicholas flamel', 'enchantress the secrets of the immortal nicholas flamel']''
**Observation**: Although the model managed to generate 5 books that map to KITAB's author book list, no titles satisfy the word-count constraint.

---

**[EXAMPLE 2]: Remaining high unsatisfaction rate with CONTEXT.**

**Author**: Gabriel García Márquez
**Constraint**: Book title ends with the letter a.
**KITAB ground truth**: ``['Love in the time of cholera', 'The fragrance of guava']''
**KITAB all books (context)**: ``['100 Years of Solitude (1967)', 'Armando Morales (2004)', 'Autumn of the Patriarch (2008)', 'Chronicle of a Death Foretold (1981)', 'Clandestine in Chile (1985)', 'Collected Novellas (1990)', 'Conversations with Gabriel Garcia Marquez (2005)', 'For the sake of a country within reach of the children (1996)', 'Gabriel Garcia Marquez (2015)', 'Granta 31 (1990)', 'Harvard Diss Romance Set (1942)', "I'm Not Here to Give a Speech (2014)", 'In evil hour (1962)', 'Innocent Erendira and Other Stories (1979)', 'Leaf storm (1955)', 'Living to tell the tale (2002)', 'Love in the time of cholera (1985)', 'Memories of My Melancholy Whores (2004)', 'News of a kidnapping (1996)', 'No One Writes to the Colonel (1961)', 'Of love and other demons (1994)', 'One hundred years of solitude (1967)', 'Scandal of the Century (2020)', 'Strange Pilgrims (1992)', 'The Handsomest Drowned Man in the World (1968)', 'The Short story (1987)', 'The fragrance of guava (1982)', 'The general in his labyrinth (1989)', 'The story of a shipwrecked sailor (1970)', 'Three Novellas (1991)', 'Tomas Sanchez (2003)']''
**GPT4 books**: ``['in evil hour', 'of love and other demons']''
**Observation**: Although the model successfully follows the provided context and all books are in fact by the author, it fails at extracting books satisfying the constraint.

---

---

**[EXAMPLE 3]: SELF-CONTEXT leading to higher information irrelevance.**

**Author**: Russell Humke Fitzgibbon
**Constraint**: Book title starts with the letter v.
**KITAB all books for the author**: ```['A directory of Latin American political parties (1970)', 'Assessing democracy in Latin America (1998)', 'Brazil: a chronology and fact book, 1488-1973 (1974)', 'Cuba and the United States, 1900-1935 (1935)', 'Global politics (1944)', 'Latin America (1946)', 'The constitutions of the Americas, as of January 1, 1948. (1948)', 'Uruguay (1954)', 'Visual outline of Latin American history (1938)']``
**GPT4 self-context**: ```['vatican and the kremlin', 'vatican in world politics', 'catholic church and the authoritarian regime in austria', 'catholic church and the totalitarian regime in italy', 'catholic church and the dictatorship in portugal', 'catholic church and the fascist regime in spain']``
**Observation**: None of the books in the model output are from the author. The model starts by generating two irrelevant books that do satisfy the constraint but are not from the author. Given that context, it then continues to autoregressively generate more of the same books on similar topics (religion) but that are still irrelevant and also do not satisfy the constraint anymore.

---

**[EXAMPLE 4]: SELF-CONTEXT leading to higher information irrelevance.**

**Author**: Johnston McCulley
**Constraint**: Book contains a human name in its title.
**KITAB all books for the author**: ```['Alias the thunderbolt (1927)', 'Avenging Twins Collect (2007)', 'Black Grandee (2019)', 'Brand of Silence (2020)', 'Broadway Bab (1919)', 'Bullet law (1960)', 'Canyon of peril (1935)', 'Captain Fly-By-Night (2012)', 'Companions of Jehu (2008)', 'Demon (2007)', "Devil's Doubloons (2019)", 'Don Renegade (2017)', 'Ghost Phone (2014)', 'Iron Horse Town.  (1952)', "Johnston McCulley's Zorro (2000)", 'King of Chaos and Other Adventures (2019)', 'Land of Lost Hope (2016)', 'Man Who Changed Rooms (2008)', 'Scarlet Scourge (2007)', 'Slave of Mystery and Other Tales of Suspense from the Pulps (2006)', 'South of the pass (1944)', 'Spider Strain (2017)', 'Tales of Thubway Tham (2005)', 'Texas showdown (1953)', "A white man's chance (1927)", 'The Black Star (1921)', 'The Cougar Kid (1945)', 'The Rollicking rogue (1941)', 'The blocked trail (1932)', 'The caballero. (1947)', 'The mark of Zorro (1924)', 'The masked woman (1920)', 'The range cavalier (1933)', "The rangers' code (1924)", 'Three Mysteries by Johnston Mcculley (2009)', 'Tragedy Trail (2020)', 'Western Megapack (2011)', 'Wild Noreen (2007)', "Women's War (2008)"]``
**GPT4 self-context**: ```['mark of zorro', 'bold caballero', 'crimson clown', 'crimson clown again', 'crimson clowns apprehension', 'crimson clowns return', 'crimson clowns success', 'crimson clowns triumph', 'crimson clowns victory', 'crimson clowns willingness', 'crimson clowns yielding', 'crimson clowns zeal', 'crimson clowns zest', 'crimson clowns zestfulness', 'crimson clowns zestiness', 'crimson clowns zestlessness', 'crimson clowns zestlessnesslessness', 'crimson clowns zestlessnesslessnesslessness', 'crimson clowns zestlessnesslessnesslessnesslessness', ....]``
**Observation**: The first 3 books returned by the model for context are in fact by the author (and more or less satisfying the constraint), however, the model continues to autoregressively generate irrelevant and hallucinated title variations from the 'Crimson Clown' in a continuous problematic pattern.

