# OpenReview forum: "KITAB: Evaluating LLMs on Constraint Satisfaction for Information Retrieval"
_ICLR.cc/2024/Conference — ICLR 2024 poster_

### Official Review · Reviewer_dWav · 2023-10-30

**Soundness:** 2 fair
**Presentation:** 3 good
**Contribution:** 4 excellent
**Rating:** 8
**Confidence:** 3

**Summary:**

This paper develops a dataset and associated dataset construction framework for the evaluation of LLMs on _constraint satisfaction information retrieval_ tasks (eg, "celebrities who died in 2022"). This problem is interesting because many it covers many important and useful factual query tasks, but LLMs have significant accuracy difficulties. The authors experiment against existing LLMs (GPT-3.5 and GPT-4) while varying key dimensions of the task, and conduct performance and error analyses on the results.

**Strengths:**

A crisp task definition and associated evaluation dataset can be a high-leverage research contribution that enables and accelerates other work, so this is a worthwhile goal. It's also quite promising that the task seems to be intuitively straightforward yet is still challenging for state-of-the-art proprietary models like GPT-4.

The citations and engagement with IR research going back to 1989, 1999, 2009, etc is good to see and contextualizes the work well.

The exploration of problem setting parameters (task settings, author popularity, selectivity) is systematic and illuminating. The research questions are clearly defined and answered. The explanations of the dataset development and scoring are clear and thorough. The "extensible" nature of the methodology for dataset construction is useful to "future proof" the work.

Some of the detailed error analysis was intriguing and suggestive of mechanisms behind the failure modes which is very exciting for future work:
* ends-with vs starts-with constraints
* "sharp phase transition" for failure rate vs author popularity
* fabrication for "list all books by author"

**Weaknesses:**

For the open datasets (Open Library and WikiData), is there some precautions taken to "snapshot" these to a specific point in time for reproducibility purposes? Also, the procedures uses a few proprietary tools to develop the dataset: Azure Cognitive Services NER and Language API, Geonames in the dataset preparation. If the framework code release is open-sourced, future developers could substitute open or more reproducible components in these places, but this means that it would be difficult to create a new version of the dataset "out of the box". Likewise for GPT-3.5 and -4: are all the queries and responses captured somehow for reproducibility of the experiments and analysis? Some of these reproducibility concerns potentially undercut the usefulness and extensibility of the framework.

For controlled experimentation purposes it is useful to focus on a single domain such as book  authorship. However, it seems like there is some risk that the findings or behaviors may not transfer to other constrained IR tasks. How easy or difficult would it be to adapt the framework to generate similar dataset-task pairings in other domains as mentioned in the paper (movies, restaurants, etc)?

Can fine-tuned LLMs do well on KITAB? The results and dataset would be more compelling with another FINE-TUNED setting, even if it had to be done with an OSS model.

**Questions:**

What does KITAB stand for? I couldn't find a definition.

Table 3: why are some numbers red?

---

> ### Author Response · Authors · 2023-11-15
> **Response to Reviewer dWav: Part 1 of 2**
>
> *[part 1 of 2]*
>
> We thank the reviewers for their detailed feedback and for finding our work to be a valuable worthwhile contribution. Next, we respond to each of the comments and questions. We’d also be happy to follow up with further clarifications.
>
> **For the open datasets (Open Library and WikiData), is there some precautions taken to "snapshot" these to a specific point in time for reproducibility purposes?** - Indeed, for the sake of reproducibility, we follow two key precautions:
> - *Inclusion of Raw Data and Metadata*: In our published dataset, in addition to our clean processed book data associated with each query, we have included the raw snapshot pulled from OpenLibrary. We have also included the metadata (birth year and popularity metric) associated with each author. This is integrated alongside the main KITAB dataset.
> - *Provision of Code Scripts*: We provide code scripts not only for the evaluation of KITAB, but also for every step of the construction of the dataset itself, allowing users to refresh and rebuild the dataset as needed and also generate a different version with a different set of authors if necessary.
>
> **The procedures uses a few proprietary tools to develop the dataset: Azure Cognitive Services NER and Language API, Geonames in the dataset preparation.** - We experimented with a few options to identify an effective and inclusive entity recognition tool and found that Azure Cognitive Services provided the most reliable and comprehensive results for our dataset. However, if users would like to replicate or update KITAB, and would like to adhere to non-proprietary tools, they can easily adapt our published code scripts to replace Azure's API with any open-source NER software they see fit. Additionally, we used Geonames, which is already an open-source tool, making it readily accessible for those who wish to use or adapt our methodology. For reference, you can check out Geonames - All Cities with a population > 1000 — Opendatasoft.
>
> **Likewise for GPT-3.5 and -4: are all the queries and responses captured somehow for reproducibility of the experiments and analysis? Some of these reproducibility concerns potentially undercut the usefulness and extensibility of the framework.** -
> This feedback is very insightful; we have opted not to release our model responses along with KITAB, primarily due to the dynamic nature of LLMs, especially GPT models and the inconsistencies that can arise from different endpoints, leading to potential confusion. However, we have thoroughly documented all of our experiments, capturing both the raw and processed outputs from the models. To facilitate research and extend the utility of our work, we plan to include a note in our published dataset, encouraging researchers to contact the authors directly if they wish to access these detailed records. This approach is intended to maintain the integrity and reproducibility of our research while acknowledging the evolving landscape of LLMs.

---

> > ### Author Response · Authors · 2023-11-15
> > **Response to Reviewer dWav: Part 2 of 2**
> >
> > *[part 2 of 2]*
> >
> > **For controlled experimentation purposes it is useful to focus on a single domain such as book authorship. However, it seems like there is some risk that the findings or behaviors may not transfer to other constrained IR tasks. How easy or difficult would it be to adapt the framework to generate similar dataset-task pairings in other domains as mentioned in the paper (movies, restaurants, etc)?** - As noted in the general response, we chose the book domain as a domain that has sufficient public data to work with, but we believe that the problem of LLMs having difficulties in answering such queries is more ubiquitous. Based on research and usage evidence, similar issues have been observed when models are required to answer queries of the same nature in the scientific research domain or encyclopedia-like questions by models fabricating titles of papers that do not exist on a given topic (e.g. https://twitter.com/IttefaqM/status/1715392338535563594?s=20). Most prominently, even the earlier hallucination observed in Bard (see https://www.theverge.com/2023/2/8/23590864/google-ai-chatbot-bard-mistake-error-exoplanet-demo) was related to a similar query that also contained constraints which were not satisfied by the model (i.e., “What new discoveries from the James Webb Space Telescope can I tell my 9 year old about?” with the model including a discovery that did *not* actually involve the James Webb Space Telescope). In addition, we also wanted to provide future users with a way to create more evaluation data dynamically by using public APIs, given rising concerns around data contamination.
> > The level of difficulty associated with constructing datasets in a similar vein under other domains is related to data cleaning and completeness. This process needs to be customized. Even for domains around movies, restaurants, celebrities etc., well-known repositories are far from clean and require the integration of several data sources and robust cleaning approaches.
> > While we hope that more datasets of a similar nature evolve in the community covering other domains, we hope that KITAB opens the path for such work and shows the value of being able to collect and leverage clean and complete data for similar evaluations. In addition, one can utilize and adapt our construction framework discussed in the paper and supplementary material as well as our comprehensive code scripts.
> >
> > **Can fine-tuned LLMs do well on KITAB? The results and dataset would be more compelling with another FINE-TUNED setting, even if it had to be done with an OSS model** - This is indeed a promising direction. While our experiments have primarily been with general-purpose models and haven't extensively explored models fine-tuned for specific tasks like constraint satisfaction, KITAB offers a valuable opportunity for such experimentation, in particular when seen as a dynamic data collection approach which we facilitate with the release of the data collection workflow code. It would be interesting to see how instruction-tuned models perform in this setting and how KITAB performance on these models aligns with general IR performance. To this end however, we believe that fine-tuning or adaptation for constraint satisfaction is a basic skill that all pretrained models need to have to enable controlled generation and that it is not a task for which every user or customer should have to fine-tune for.
> >
> > **What does KITAB stand for? I couldn't find a definition.** - The dataset is named after the word kitab (https://en.wikipedia.org/wiki/Kitab), which is the word for "book" in Arabic, Swahili, Urdu, Hindi and various Indian and Turkic languages.
> >
> > **Table 3: why are some numbers red?** - We distinguish some of the results in red to highlight the lowest performing constraint type per experimental condition/evaluation metric (column). This is merely for the sake of improved readability.

---

> > > ### Comment · Reviewer_dWav · 2023-11-18
> > > **Author response**
> > >
> > > Thank you for the response. The additional steps around reproducibility address some of my concerns on that front.
> > >
> > > It would be nice to put a footnote about the origin/definition of kitab :)

---

### Official Review · Reviewer_XGyb · 2023-11-01

**Soundness:** 3 good
**Presentation:** 3 good
**Contribution:** 3 good
**Rating:** 6
**Confidence:** 3

**Summary:**

This paper introduces a new dataset, KITAB that evaluates how well LLMs  perform in answering constraint satisfaction queries for information retrieval.

**Strengths:**

Overall I think this new dataset is a well posed and timely work.

* The dataset seems well constructed and reasonable
* Incorporating constraint satisfaction for LLMs appears to be somewhat understudied, and this dataset will fit nicely into the existing LLM evaluation framework
* The choice of model evaluation on the data (ChatGPTs) is reasonable
* The differing frames for the data (with context, without, etc) makes sense
* The dataset construction is well documented, with design choices explicitly described
* The metrics and evaluations are well constructed

**Weaknesses:**

The dataset is somewhat limited, being about books, years and authors. I would prefer if the dataset included a wider portfolio of constraint tasks, perhaps around geography or movies.

It would also be nice if at least one opensource LLM was used in evaluation (i.e. LLAMA or LLAMA 2).

Some of the constraints seem a little artificial: authors born in a specific year. Some of the constraint questions are out of scope of the design of current LLM's tokenization, "Book title starts with the letter v". These constraints don't accurately reflect user interaction with a LLM, I feel.

**Questions:**

Can you enumerate all the constraints? It's not clear from the text what exactly the single and double constraints general architecture is.

---

> ### Author Response · Authors · 2023-11-15
> **Response to Reviewer XGyb: Part 1 of 2**
>
> *[part 1 of 2]*
>
> We thank the reviewers for their elaborate feedback and for finding our work to be well-constructed and a good fit. Next, we respond to each of the comments and questions. We’d also be happy to follow up with further clarifications.
>
> **The dataset is somewhat limited, being about books, years and authors. I would prefer if the dataset included a wider portfolio of constraint tasks, perhaps around geography or movies.** - As mentioned in the general response, we chose the book domain as a domain that has sufficient public data to work with, but we believe that the problem of LLMs having difficulties in answering such queries is more ubiquitous. Based on research and usage evidence, similar issues have been observed when models are required to answer queries of the same nature in the scientific research domain or encyclopedia-like questions by models fabricating titles of papers that do not exist on a given topic (e.g. https://twitter.com/IttefaqM/status/1715392338535563594?s=20). Most prominently, even the earlier hallucination observed in Bard (see https://www.theverge.com/2023/2/8/23590864/google-ai-chatbot-bard-mistake-error-exoplanet-demo) was related to a similar query that also contained constraints which were not satisfied by the model (i.e., “What new discoveries from the James Webb Space Telescope can I tell my 9 year old about?” with the model including a discovery that did *not* actually involve the James Webb Space Telescope). While we hope that more datasets of a similar nature evolve in the community covering other domains, we hope that KITAB opens the path for such work and shows the value of being able to collect and leverage clean and complete data for similar evaluations. Note that one of the main challenges with constructing datasets in a similar vein is related to data cleaning and completeness. Even for domains around movies, geography, celebrities etc., well-known repositories are far from clean and require the integration of several data sources and robust cleaning approaches as we further discuss in the paper and supplementary material.  In addition, we also wanted to provide future users with a way to create more evaluation data dynamically by using public APIs, given rising concerns around data contamination.
>
> **It would also be nice if at least one opensource LLM was used in evaluation (i.e. LLAMA or LLAMA 2)** -  This is a very important point and we very much agree with the reviewer’s observation. Unfortunately, at the time of investigation models in the llama family (including chat versions) did not perform well in terms of following formatting instructions, which resulted in challenges with reliable and scalable evaluation. More recently, and in particular the official deployments by Meta, are more receptive to formatting instructions, which we hope will enable us and future studies to overcome the challenges that we faced initially. At the same time, even with the most recent deployments and releases, we still notice that for this particular task model quality is so low that it is not clear whether it is possible to extract useful quantitative or qualitative insights, in particular when it comes to comparing many different open source models or performance on different constraint types. This concern was valid even for the most popular authors in the dataset, but for the mid-range and less popular authors anecdotally we see that model output may not even have a single satisfactory item. We will include a discussion on this very important topic in the paper and hope to do more of these tasks in the future for either more simplified versions of our dataset or for future open source model releases that perhaps show stronger emerging capabilities.

---

> ### Author Response · Authors · 2023-11-15
> **Response to Reviewer XGyb: Part 2 of 2**
>
> *[part 2 of 2]*
>
> **Some of the constraints seem a little artificial: authors born in a specific year. Some of the constraint questions are out of scope of the design of current LLM's tokenization, "Book title starts with the letter v". These constraints don't accurately reflect user interaction with a LLM, I feel.** - Your observation about certain constraints seeming artificial, like specifying authors' birth years and questions about book titles starting with a specific letter, is indeed insightful. The reasoning we’ve arrived at behind these constraints, however, comes from a practical standpoint.
> As multiple authors could share the same name, we have chosen to add the birth year in our prompts to help in distinguishing them. This approach is aimed at reducing confusion and improving the accuracy of responses, especially in cases where authors might be less well-known or have common names. We indeed saw this issue happening in practice and merely wanted to add more information to help the model and provide a better chance for it to be correct as it did in fact help for several examples that we investigated manually.
> Regarding constraints such as book titles starting with a specific letter, while it is indeed valid that these type of tokenization-dependent tasks might be out of scope for the LLM's architecture, we have decided to include them to assess the model's performance under various scenarios, keeping in mind that average users may not be aware of the technical limitations of language models and might still trust or expect accurate responses from the model in such cases. Another consideration is that currently users are building applications and multi-agent systems on top of existing LLMs and they are using them for tasks that traditionally we had not anticipated them for, even searching with a given string (or letter). While current LLMs do not perform well on such tasks, even if they are out-of-scope for their design, it is probably still a good idea to quantify and increase awareness as results remain unreliable.
>
>  **Can you enumerate all the constraints? It's not clear from the text what exactly the single and double constraints general architecture is** - Certainly, we used two main types of constraints in our study:
> - Author Constraints: This specifies the author's name and their birth year. For example, "books by Gabriel García Márquez (born 1927)".
> - Book Constraints: These are of six different types (mentioned in Table 2) to test for lexical, entity-based, and temporal aspects. An example is "Book contains a human name in its title". Please refer to Table 2 for details on the distribution of book constraints of different types across KITAB queries containing one and two book constraints (in addition to an author constraint), as well as Figures 5 and 6 for a visualization of the latter across author popularity and constrainedness.
>
> In our experiments, for every query, we use one author constraint and a varied number of book constraints from zero to two. We then evaluate how well the models met all these constraints per query through a variety of metrics described in detail in section 4.1.

---

### Official Review · Reviewer_ZHUe · 2023-11-04

**Soundness:** 3 good
**Presentation:** 4 excellent
**Contribution:** 3 good
**Rating:** 6
**Confidence:** 3

**Summary:**

The paper proposes KITAB, a dataset for evaluating how large language models perform on information retrieval tasks with constraint satisfaction. The author(s) describe the dataset constructions and settings in detail. Evaluated on the proposed dataset, observations from different perspectives indicate that KITAB is still a challenging task for LLMs.

**Strengths:**

1. Timely study of LLMs on constraint satisfaction for information retrieval.
2. The overall paper is well-written and easy to follow.
3. Detailed descriptions of the data construction are provided.
4. The author(s) also promise to release the dataset, as well as code to the community.
5. The author(s) comprehensively analyzed the results to draw several interesting observations.

**Weaknesses:**

1. Concerns of the submission being out-of-scope. The paper undoubtedly stands out as a great resource that makes contributions through its new dataset and the accompanying empirical studies. It certainly serves as a valuable asset for the research community. However, I'm concerned about its suitability as a full paper for ICLR, given the conference's typical focus.

2. The author(s) have indeed conducted a thorough consideration of multiple facets in constructing the dataset, such as the number of constraints, the variety of constraints, and unsatisfiability. Nevertheless, the scope of the proposed dataset is fundamentally narrow, being confined to the domain of books. This domain-specific focus could limit its applicability to a broader range of IR research. It could be better if data from a variety of domains can be included.

3. The evaluations presented are primarily focused on the LLM services provided by OpenAI. While these are undoubtedly among the leading services in the field, the inclusion of open-source LLMs, such as LLaMA 2 chat, in the evaluation process could provide a more comprehensive view of the landscape.

**Questions:**

Please refer to "Weaknesses".

---

> ### Author Response · Authors · 2023-11-15
> **Response for Reviewer ZHUe**
>
> We thank the reviewer for finding our study timely, comprehensive, and well presented!  Next, we respond to each of the comments and questions. We’d also be happy to follow up with further clarifications.
>
> **Venue fit. Concerns of the submission being out-of-scope.** - It would be great if the reviewer could further clarify their concerns on scope. If the concern is related to the fact that the dataset focuses on factual IR queries rather than learning problems, it is worth noting that even though IR queries traditionally have not been considered in the ML scope, given the increasing abilities and most importantly usage of LLMs to answer these queries, it is important to share these results and artifacts with the ML community through this ICLR track focused on benchmarks and datasets. Indeed, pure LLM or LLM-based solutions are currently being used worldwide to answer exactly these types of queries, even more so in the retrieval augmented generation setting (BingChat and ChatGPT more recently), which we simulate in one of our experimental conditions.
>
> **Limited Scope. … The scope of the proposed dataset is fundamentally narrow, being confined to the domain of books.** - As mentioned in the general response, we chose the book domain as a domain that has sufficient public data to work with, but we believe that the problem of LLMs having difficulties in answering such queries is more ubiquitous. Based on research and usage evidence, similar issues have been observed when models are required to answer queries of the same nature in the scientific research domain or encyclopedia-like questions by models fabricating titles of papers that do not exist on a given topic (e.g. https://twitter.com/IttefaqM/status/1715392338535563594?s=20). Most prominently, even the earlier hallucination observed in Bard (see https://www.theverge.com/2023/2/8/23590864/google-ai-chatbot-bard-mistake-error-exoplanet-demo) was related to a similar query that also contained constraints which were not satisfied by the model (i.e., “What new discoveries from the James Webb Space Telescope can I tell my 9 year old about?” with the model including a discovery that did *not* actually involve the James Webb Space Telescope). While we hope that more datasets of a similar nature evolve in the community, we hope that KITAB paves the path for such work and shows the value of being able to collect and leverage clean and complete data for similar evaluations. Note that one of the main challenges with constructing datasets in a similar vein is related to data cleaning and completeness. Even for domains like movies, celebrities etc., well-known repositories are far from clean and require the integration of several data sources and robust cleaning approaches as we further discuss in the paper and supplementary material. In addition, we also wanted to provide future users with a way to create more evaluation data dynamically by using public APIs, given rising concerns around data contamination.
>
> **Inclusion of Open Source Models** - This is a very important point and we very much agree with the reviewer’s observation. Unfortunately, at the time of investigation models in the llama family (including chat versions) did not perform well in terms of following formatting instructions, which resulted in challenges with reliable and scalable evaluation. More recently, and in particular the official deployments by Meta, are more receptive to formatting instructions, which we hope will enable us and future studies to overcome the challenges that we faced initially. At the same time, even with the most recent deployments and releases, we still notice that for this particular task model quality is so low that it is not clear whether it is possible to extract useful quantitative or qualitative insights, in particular when it comes to comparing many different open source models or performance on different constraint types. This concern was valid even for the most popular authors in the dataset, but for the mid-range and less popular authors anecdotally we see that model output may not even have a single satisfactory item. We will include a discussion on this very important topic in the paper and hope to do more of these tasks in the future for either more simplified versions of our dataset or for future open source model releases that perhaps show stronger emerging capabilities.

---

> > ### Comment · Reviewer_ZHUe · 2023-11-22
> >
> > Thank you for your comprehensive and thoughtful responses to my concerns. I appreciate your time and effort. Overall, I will maintain my rating.
> >
> > For the weakness "Venue fit. Concerns of the submission being out-of-scope.", thank you for your detailed response clarifying the scope and relevance of your submission to ICLR. I realize now that the ICLR this year includes a specific track for benchmarks and datasets, which indeed makes your paper a suitable fit for the conference. I apologize for my oversight in this regard.

---

### Official Review · Reviewer_pbw5 · 2023-11-08

**Soundness:** 3 good
**Presentation:** 3 good
**Contribution:** 2 fair
**Rating:** 5
**Confidence:** 4

**Summary:**

This paper proposes a dataset KITAB for testing on how LLMs can satisfy constraints in IR. A set of books are collected. Constraints can be set on authors, titles, dates, etc. and can be combined. The evaluation of the results by GPT-3.5 and GPT-4 are evaluated in terms of irrelevance, completeness, etc. Three test conditions are set using different prompts: no-context, self-context and context, providing (or not) the list of books written by an author.
The paper examines how GPT-3.5 and GPT-4 perform on different queries. A wide range of experimental results are reported.

**Strengths:**

The problem examined - constraint satisfaction - is not a widely studied problem. To some extent, this investigation extends the traditional IR to a special type of IR.
The data collection and experimental setting are well described.
The results may reveal that GPT models are not able to process this type of queries very well, and in particular, those with some constraints.
The authors will release the dataset.

**Weaknesses:**

The targeted problem is very special. The authors consider it as an IR problem. It resemble more database querying problem. Indeed, once the book references are transformed into structured database, the problem may be much simpler. Howere, as I understand, the intent of the author is not to examine how GPT models can handle such a search problem, but try to use this to test the ability of LLMs to handle queries containing constraints. While I appreciate this effort, I still question on the appropriateness of thFor e task. Instead of expression general constraints in a language, the constraints examined are very specific, and sometimes special. For example, constraints asking a title to end with a letter `v', or titles to be of 4 words, seem very particular. It is expected that a general LLM may not perform the task well because they have not been trained for the task. The main concern about the work is whether the types of constraints used can really test the ability of LLMs to satisfy constraints, or if the tests are on LLMs ability to understand the constraints and to execute filterings of books accordingly in this very special case. Even if LLMs are able to do the job well, would one be able to draw interesting conclusions? The experiments shown in the paper may not allow to draw general conclusions on LLM's ability to satisfy constraints.
Given the very special characteristics of the dataset, I question about the value of the KITAB dataset for the purpose of examining LLM's capability of constraint satisfaction in search.
While the ground truth is determined correctly, there may be some problem in the measure of irrelevance and completeness, as some fuzzy matches are allows. As the authors admit, there may be some overestimation of the performance. So the experimental results are only indicative, with some possible rate of errors.

**Questions:**

For the estimation of irrelevance or partial satisfaction, one case of match is when one is a string subset of another: For each ki, we check if there exists a book in the ground truth set of books by that author which is either a string subset match for ki (in both directions),... Do you also apply a threshold on the percentage of the substring, or a substring of any length is accepted?

---

> ### Author Response · Authors · 2023-11-15
> **Response for Reviewer pbw5**
>
> We thank the reviewer for their feedback and for appreciating our contributions! Next, we respond to each of the comments and questions. We’d also be happy to follow up with further clarifications.
>
> **Generality of the dataset. Can the presented constraint types really test the ability of LLMs to satisfy constraints?** - We consider 6 different constraint types to test for *lexical*, *entity-based*, and *temporal* constraints. Indeed, the pretrained models are not directly trained to satisfy these constraints, but that is the case for many emergent abilities of interest. Yet, we’d like to understand and measure model performance on these abilities given that current models do show non-trivial (albeit non satisfactory) performance in such tasks.
>
> Even amongst the constraint types we consider, models may have received direct or indirect training signals. For example, entity recognition (human/city name) is a task that has been largely studied in NLP and may therefore have more explicit training data used for pretraining, while the model’s ability to apply string operations is less studied and perhaps requires different skills. This is why we disaggregate our analysis and show how performance varies across different constraints. While it is true that some of these constraints can be easily satisfied through SQL, our goal here is to test the model’s direct linguistic abilities as at the application-level not all user queries are or can be rewritten as SQL queries. Some of the constraint types we consider, can actually be mapped to concrete *librarian tasks* such as retrieving an author’s work (List all books by an author), indexing or searching by a given constraint (List all books by an author whose title starts with a letter), finding characters mentioned in the title (or content in more advanced versions) etc. While our goal isn’t to completely solve the library domain, we use this domain as an illustration of a much larger problem in factual inaccuracy when models are not able to follow constraints.
>
> **Evaluation procedure. While the ground truth is determined correctly, there may be some problem in the measure of irrelevance and completeness, as some fuzzy matches are allowed.** - Thanks for raising this concern, it is indeed important for the measurement to be realistic. We view performance on KITAB as an “upper bound” on the LLM’s actual ability for these types of simple information retrieval tasks. Our main qualitative result is that existing LLMs already struggle to obtain good performance on KITAB, and therefore, this casts doubt on their ability to perform these sorts of constraint satisfaction tasks in general settings. This is why we are willing to overestimate model performance; indeed, our main result demonstrates that even with all of these relaxations, state-of-the art LLMs still struggle immensely with this elementary (but important) task.
>
> In addition, relaxing the evaluation process is crucial for testing constraint satisfaction abilities in the language domain in a realistic manner without undermining model capabilities. For example, the evaluation should not over penalize the model if the model is asked to output a title that starts with the letter “c” and includes the title “The Citadel” in its results. Indeed, librarian practices do include such titles as this categorization is most useful for users. Similarly, if the model outputs the following title “Gödel, Escher, Bach” as a book by Douglas R. Hofstadter but does not mention its full title (“Gödel, Escher, Bach: An Eternal Golden Braid”) this should be considered a correct answer as the book is known with both titles. Similar reasoning goes to accounting for potential typos in the model’s answer or in our data by using fuzzy matching. In fact, we consider the evaluation process together with the appropriate implemented relaxation process as a practical contribution of the paper and will open source the artifacts to support reproducibility to this end.
>
> As mentioned in the paper (Section 3.1 - Book collection) and supplementary material (C.1), we also run a manual annotation to separately evaluate how irrelevance and completeness is impacted by perhaps books that we were not able to retrieve with the sources used (OpenLibrary and WikiData) and find that only 5% of the GPT4 answers contain at least one book that exists on the internet from the same author but is not captured from KITAB. The impact of this on actual irrelevance and completeness is a lot lower because the number of non-found books per query is usually less than 2 while the satisfactory set is larger.
>
> **Do you also apply a threshold on the percentage of the substring, or a substring of any length is accepted?** - Let’s take the example of 2 possible titles for the same book t1= “Gödel, Escher, Bach: An Eternal Golden Braid” and t2= “Gödel, Escher, Bach”. We would consider this a match if t2 \in t1 or if t1 \in t2. Therefore, we only consider full substrings as a match.

---

> > ### Comment · Reviewer_pbw5 · 2023-11-22
> >
> > Thanks for the explanation and answer.
> >
> > The task still seems very special, which may not be well suited for testing the general ability of LLM for constraint satisfaction.
> > The task is more suitable for SQL-like querying system than a LLM.
> >
> > Despite the fact that 6 types of constraints are considered, they are very specific for the task. Therefore, their representativity of the general constraints is limited.

---

> > > ### Author Response · Authors · 2023-11-22
> > > **Generality of the dataset**
> > >
> > > Thanks for engaging in the discussion!
> > >
> > > Even though KITAB only focuses on the literature domain, it is a suitable benchmark for studying the more general and widespread issue by offering a structured way of verifying information. We hope that future efforts will extend this to further domains.
> > >
> > > While many of our queries can be casted to SQL and tested on structured information, we'd like to note that information is not always structured or even available as such. That is why users are indeed using systems like BingChat, Bard, and ChatGPT to ask such questions.
> > >
> > > Similar issues to what we present in KITAB have been observed when models are required to answer queries of the same nature in the scientific research domain or encyclopedia-like questions by models fabricating titles of papers that do not exist on a given topic (e.g. https://twitter.com/IttefaqM/status/1715392338535563594?s=20). Most prominently, even the earlier hallucination observed in Bard (see https://www.theverge.com/2023/2/8/23590864/google-ai-chatbot-bard-mistake-error-exoplanet-demo) was related to a similar query that also contained constraints which were not satisfied by the model (i.e., “What new discoveries from the James Webb Space Telescope can I tell my 9 year old about?” with the model including a discovery that did not actually involve the James Webb Space Telescope).

---

### Author Response · Authors · 2023-11-15
**General Response**

We thank the reviewers for their in-depth comments. We would like to first address a concern shared by many of the reviewers in addition to individual responses to each reviewer.

**Generality of the dataset**: A number of reviewers express concern about the breadth of the dataset. We wish to emphasize that our interest in the KITAB dataset is to use it as a representative task for commonplace information retrieval-style queries that LLMs are increasingly used for in popular deployments such as Bing Chat and Google Bard. We chose to focus on the domain of books for this purpose for two reasons. First, existing APIs allowed us to automatically generate and verify a large number of the types of query constraints that we wanted. Secondly, APIs also allow other researchers to generate a fresh copy of KITAB, which is very important for future evaluations in light of the risk of data contamination for any fixed, static, and saturated dataset.

It is an interesting question to create an even larger-scale dataset for these types of queries, however, we do not know of any methodology for generating a large, future-proof dataset like KITAB for general domains. We also do not have any reason to believe why performance in the domain of books is not representative of general performance in other domains; we remark that similar phenomena to those that we have uncovered have also been documented in other settings such as when models fabricate paper titles or events. The main advantage of KITAB to this prior work is the scale and the dynamic nature of the dataset, which allows us to make more quantitative measurements of these phenomena. We note that this scale is enabled exactly because we choose to use the specialized APIs in the domain of books.

We also note that this is why we believe it is most interesting to evaluate large, general-purpose LLMs on KITAB, as these are the sorts of models used in the types of applications mentioned above.

**Generality of the task**: A number of reviewers are also concerned about the utility of the information retrieval-style queries that we consider. In fact, simple IR-style queries with constraints like those in KITAB are natural use-cases for commercial deployments of LLMs in search engines like Bing Chat or Google Bard. Therefore, the fact that state-of-the-art models fail to perform well on KITAB demonstrates a crucial weakness of these existing applications. We consider 6 different constraint types to test for *lexical*, *entity-based*, and *temporal* constraints for the domain, and in many cases these constraints map to tasks that librarians or users would perform such as indexing, search, retrieving an author’s work etc.

**Evaluation on other opensource models**: This is a very important point, and we very much agree with the reviewer’s observation. Unfortunately, at the time of investigation models in the llama family (including chat versions) did not perform well in terms of following formatting instructions, which resulted in challenges with reliable and scalable evaluation. More recently, and in particular the official deployments by Meta, are more receptive to formatting instructions, which we hope will enable us and future studies to overcome the challenges that we faced initially. At the same time, even with the most recent deployments and releases of open-source models, we still notice that for this particular task model quality is so low that it is not clear whether it is possible to extract useful quantitative or qualitative insights, in particular when it comes to comparing many different open source models or performance on different constraint types. This concern was valid even for the most popular authors in the dataset, but for the mid-range and less popular authors anecdotally we see that model output may not even have a single satisfactory item. We will include a discussion on this very important topic in the paper and hope to do more of these tasks in the future for either more simplified versions of our dataset or for future open source model releases that perhaps show stronger emerging capabilities.

---

### Meta-Review · Area_Chair_7gXY · 2023-12-15

**Metareview:**

The paper presents a dataset titled KITAB for measuring constraint satisfaction abilities of LLMs.  The datasent contains book related data with 13k queries and 600 authors.  The authors analyze the limitations of GPT-3.5 and GPT-4 on this dataset and suggest that open sourcing this resource can help make progress in the area of complex question answering via LLMs.

Strengths:  The strengths of the paper are quite clear.  The dataset is interesting and measures the limitations of LLMs for an interesting class of prompts that could be quite important in real world scenarios.  The experimental setting is clearly described and the limitations of the above models are real--this suggests that this benchmark can be used to drive progress towards better (aligned) LLMs in the future.

Weaknesses:  I think that only using GPT-3.5 and GPT-4 as the models is limited--the authors could have considered newer open source models for the sake of reproducibility.  Other than that, I don't see many strong limitations.  There are concerns from the reviewers regarding the task being niche (which I don't agree with--there are real queries like this that people ask retrieval systems to;  increasingly we're seeing LLMs being integrated with search engines) and also not being a fit for ICLR.  I strongly disagree with the latter point as benchmarking and data are very important for the development of LLMs and we need to encourage such papers at ICLR.

**Justification For Why Not Higher Score:**

Please see some limitations above regarding the choices of LLMs used in this study.  With a broader study, this could have been a stronger paper.

**Justification For Why Not Lower Score:**

Please see above why we think the paper should be accepted.

---

### Decision · Program_Chairs · 2024-01-16

Accept (poster)